# Proteomic and phosphoproteomic analyses reveal that TORC1 is reactivated by pheromone signaling during sexual reproduction in fission yeast

**Melvin Bérard[1,2], Laura Merlini[2], Sophie G. Martin** [1,2]*

**1** Department of Fundamental Microbiology, University of Lausanne, Lausanne, Switzerland, **2** Department of Molecular and Cellular Biology, University of Geneva, Geneva, Switzerland

* Sophie.Martin@unige.ch

**Data Availability Statement:** Proteomic and phosphoproteomic datasets are available in PRIDE (project PXD056619): https://www.ebi.ac.uk/pride/

## Abstract

Starvation, which is associated with inactivation of the growth-promoting TOR complex 1 (TORC1), is a strong environmental signal for cell differentiation. In the fission yeast *Schizosaccharomyces pombe*, nitrogen starvation has distinct physiological consequences depending on the presence of mating partners. In their absence, cells enter quiescence, and TORC1 inactivation prolongs their life. In presence of compatible mates, TORC1 inactivation is essential for sexual differentiation. Gametes engage in paracrine pheromone signaling, grow towards each other, fuse to form the diploid zygote, and form resistant, haploid spore progenies. To understand the signaling changes in the proteome and phospho-proteome during sexual reproduction, we developed cell synchronization strategies and present (phospho-)proteomic data sets that dissect pheromone from starvation signals over the sexual differentiation and cell–cell fusion processes. Unexpectedly, these data sets reveal phosphorylation of ribosomal protein S6 during sexual development, which we establish requires TORC1 activity. We demonstrate that TORC1 is re-activated by pheromone signaling, in a manner that does not require autophagy. Mutants with low TORC1 re-activation exhibit compromised mating and poorly viable spores. Thus, while inactivated to initiate the mating process, TORC1 is reactivated by pheromone signaling in starved cells to support sexual reproduction.

## Introduction

All organisms are subject to variations in nutrient availability and face periods of starvation, to which they have developed adaptations ranging from arrest in a quiescent state to elaborate differentiation strategies. Differentiation responses to nitrogen (N) starvation include for instance encystation/sporulation of many protists and fungi, the transition to multicellularity in the slime mold *Dictyostelium discoideum*, the formation of appressoria for plant invasion in

archive/projects/PXD056619 A minimal dataset of imaging data is available on FigShare: https://doi.org/10.6084/m9.figshare.27756342. S1 Data provides all the numerical values underlying the graphs, except where the data is provided in S1 and S2 Tables. Uncropped western blots are provided in S1 Raw Images.

**Funding:** European Research Council Consolidator Grant (https://erc.europa.eu) CellFusion to SGM; European Research Council Advanced Grant (https://erc.europa.eu) SexYeast to SGM. The funders played no role in study design, data collection, analysis, decision to publish or preparation of the manuscript.

**Competing interests:** SGM is a member of the PLOS Biology Editorial Board.

**Abbreviations:** CV, compensation voltage; EMM, Edinburg Minimal Medium; GO, gene ontology; GPCR, G-protein-coupled receptor; HCD, higher energy collision dissociation; N, nitrogen; PC, principal component; TEAB, triethylammonium bicarbonate buffer; TMT, tandem mass tag.

fungal pathogens, or the induction of sexual differentiation in several fungi and green algae [1–5].

The presence of nitrogen is detected by signaling pathways that direct metabolic and growth responses, of which the mechanistic target of rapamycin (TOR) kinase forms a major hub. TOR kinase is part of 2 distinct complexes (TORC), of which TORC1 is a critical regulator of growth and metabolism [6,7]. TORC1 supports anabolic growth by promoting protein, nucleotide, and lipid synthesis, and represses the catabolic autophagy pathway. Thus, TORC1 inactivation upon N starvation stops growth, but also de-represses autophagy, promoting the recycling of cellular material, which allows for specific transcriptional and translational programs supporting cell differentiation. Indeed, in many organisms, TORC1 inactivation, for instance through inhibition by its specific inhibitor rapamycin, is sufficient to produce an N starvation-like response.

The fission yeast *Schizosaccharomyces pombe* is a great model to study cell differentiation in response to N starvation, as this triggers either entry into quiescence or, when compatible mates are present, sexual differentiation. Sexual differentiation depends on the master transcriptional regulator Ste11, whose function is repressed in rich medium by a host of signaling pathways, including protein kinase A, CDK1, and TORC1 [8]. Some of this regulation is known in molecular detail, with for instance the upstream Ste11 transcription factors Rst2 and Fkh2 directly phosphorylated by PKA and CDK1, respectively [9,10]. TORC1 plays a prominent role. Indeed, conditional inactivation of Tor2, the essential catalytic component of TORC1, mimics starvation responses and leads to sexual reproduction in rich media when compatible mates are present [11–14]. Inactivation of other TORC1 subunits and activators, such as Rheb or RAG GTPases, similarly leads to starvation and sexual differentiation responses upon inactivation [14–17], while TORC1 hyperactivation or loss of negative regulators represses mating [18–20]. Several additional kinases serve a positive role to induce Ste11 function and sexual differentiation, including the stress-MAPK cascade, AMPK, and TORC2 [21,22]. These pathways are strongly interlinked. For example, AMPK contributes to TORC1 inhibition upon nitrogen stress [23]; TORC1 and TORC2 function in a signaling relay to promote sexual differentiation, where inactivation of TORC1 relieves the inhibition of the phosphatase PP2A on the main TORC2 kinase effector Gad8 [24], and their function converge on downstream targets such as the SAGA transcription factor [25].

The choice between quiescence and sexual differentiation depends on the presence of compatible mates, which signal through secreted, diffusible pheromones. Two mating types, P (*h+*) and M (*h-*), each secrete their own pheromone and detect the presence of the partner pheromone by a cognate G-protein-coupled receptor (GPCR), which elicits the activation of the same Ras-MAPK cascade in both cell types [26]. Engagement of pheromone-MAPK signaling promotes the expression of Ste11, which enhances the expression of all pheromone-responsive genes, forming a positive feedback loop that locks cells in sexual differentiation. Pheromones are secreted from an initially mobile polarity patch at the cell surface and are interpreted directionally for partner cells to pair [27,28]. Enhanced pheromone perception stabilizes the patch, leading to the growth of a cell projection (called shmoo), bringing partner cells in contact. Fusion of the 2 cells to form the diploid zygote then depends on the formation of an actin fusion focus assembled by the formin Fus1, which serves to concentrate the secretion of hydrolases that digest the cell wall at the contact site [29]. The timing of fusion focus assembly and stabilization is critical to ensure that cell wall digestion is coupled to pair formation. Fusion timing is also controlled by pheromone-MAPK signaling playing a proximal role at the fusion focus, where all components of the signaling cascade accumulate [30,31]. Finally, cell–cell fusion, closely followed by karyogamy, yields the diploid zygote, which immediately suppresses mating and enters meiosis. The zygotic fate is imposed by both transcriptional and post-

transcriptional regulations involving the master meiotic regulators Mei3 and Mei2 [32,33], and culminates in the formation of four stress-resistant spores.

While many kinases (and some phosphatases) have well-established roles in promoting sexual differentiation and cell fusion, we still have limited knowledge about their substrates. Which proteins are (de-)phosphorylated to drive differentiation and the morphogenetic changes for cell projection and cell fusion? To fill this knowledge gap, we have developed protocols to synchronize cell populations during mating and conducted time-course phosphoproteomics analyses to unveil the phosphorylation changes that occur dynamically as cells differentiate and then fuse together to form the zygote. We report on the unexpected finding that pheromone signaling leads to TORC1 re-activation in conditions of N starvation and that this is necessary for efficient mating and spore formation.

## Results

### Synchronization of cell fusion by optogenetics

We aimed to describe the protein phosphorylation changes occurring during sexual differentiation and cell fusion in the fission yeast *Schizosaccharomyces pombe*. One significant issue for population-based phosphoproteomics analysis is the lack of synchrony in the differentiation process of fission yeast cells. Inspired by the tools used to synchronize the cell cycle, and for which time-course phosphoproteomics on synchronized populations have been powerful in deciphering the order of events [34], we first developed a means to synchronize cells pre-fusion and release them synchronously into the fusion process.

The Fus1 formin, which assembles the fusion focus, is a critical regulator of cell fusion, as *fus1Δ* cells are completely fusion-deficient, arresting as differentiated, paired cells. Because Fus1 is only expressed upon sexual differentiation and is solely required for cell–cell fusion [29,35], a conditional *fus1* allele would in principle allow to specifically block cells pre-fusion in restrictive conditions and synchronously promote fusion upon reactivation. Because mating is inefficient at high temperatures, we could not use temperature as the conditional change. Instead, we designed a conditional *fus1* allele (*fus1^opto^*) based on an optogenetic design, where Fus1 N- and C-termini (Fus1N and Fus1C) are expressed as distinct polypeptides respectively linked to the *A. thaliana* photosensitive CIBN and CRY2 binding partners [36] (Fig 1A). Because Fus1N lacks actin-assembly activity and Fus1C lacks localization and condensation functionalities [37,38], the 2 separate halves are predicted to be nonfunctional in the dark, leading to arrest of cell pairs pre-fusion. Blue light illumination, which promotes CRY2-CIBN binding, should reunite the 2 halves, restoring a functional Fus1, allowing synchronous entry into the fusion process.

Indeed, when *fus1^opto^* cells were starved in the dark for 6 h, they formed pre-fusion pairs, in which the CRY2-Fus1C moiety was diffusely localized (Fig 1B). Upon blue light illumination, CRY2-Fus1C immediately formed a bright focus at the site of cell–cell contact (Fig 1B) leading to cell fusion, as seen by entry of the P-cell expressed mTagBFP2 fluorophore into the M-cell. The median time between illumination and fusion was 20 min (Fig 1C). Importantly, when *fus1^opto^* cells were mated on plates kept in light conditions over 24 h, they formed zygotes as efficiently as WT cells (Fig 1D), indicating that the allele is functional in permissive conditions. As expected, expressing CRY2-Fus1C alone did not support cell fusion, but we were surprised to observe that about 20% of Fus1N-CIBN and dark-kept *fus1^opto^* cells successfully fused. These observations suggest a fusogenic activity of Fus1N, independent of actin assembly, which will be investigated elsewhere. As shown below, this weak fusogenic activity was not problematic in shorter mating reactions. Thus, the *fus1^opto^* allele provides an efficient way to block cells pre-fusion and release them synchronously into fusion.

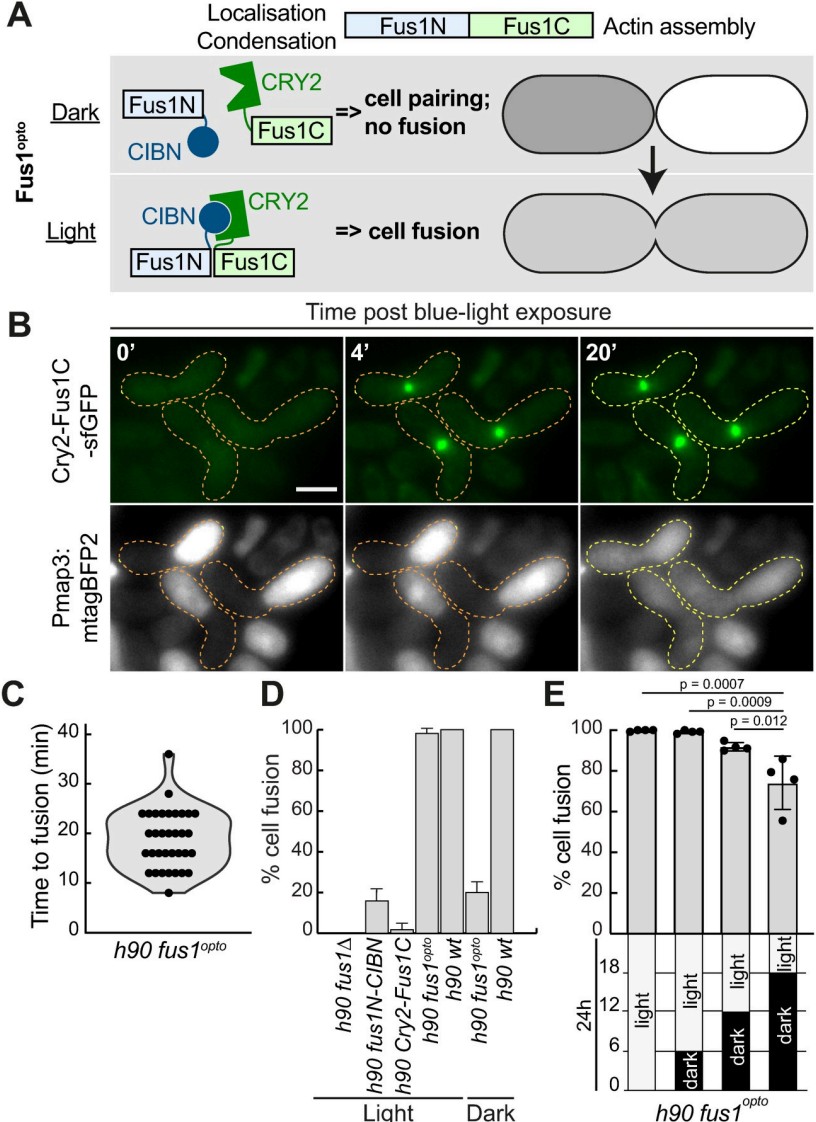

**Fig 1. Optogenetic synchronization of cell–cell fusion. (A)** Schematic representation of Fus1<sup>opto</sup> used to block cells pre-fusion in the dark and synchronize fusion upon blue light exposure. (**B**) Time-lapse movie of *h90 fus1<sup>opto</sup>* cells mated in the dark and exposed to blue light at t = 0', where Fus1C-CRY2 is tagged with sfGFP and cytosolic mTagBFP2 is expressed from the P-cell specific *map3* promoter. Dashed outlines highlight cell pairs (orange = pre-fusion pairs; yellow = fused zygotes). Scale bar = 5 μm. Original and additional data available at 10.6084/m9.figshare.27756342. (**C**) Time from light exposure to cell fusion (entry of mTagBFP2 in M-cell) in cells as in (B). *n* = 35 mating pairs quantified over 2 independent experiments. (**D**) Fusion efficiency (% of paired cells that fused) of *fus1<sup>opto</sup>* and indicated control strains after 24 h on MSL-N plates in light or dark. *N* = 3 experiments (except for WT where *N* = 2) with *n* ≥ 78 zygotes each. (**E**) Fusion efficiency of *fus1<sup>opto</sup>* after 24 h on MSL-N plates with indicated dark/light exposure over 24 h. *N* = 4 experiments with *n* ≥ 78 zygotes each. HSD Tukey *p*-values are indicated. The underlying data for panels C–E can be found in S1 Data.

In the experiment above, at the time of Fus1<sup>opto</sup> activation, only about 5% of cells were engaged in mating ([39]; see also Fig 2A, ammonium trace). In an attempt to maximize the fraction of pre-fusion cell pairs and synchronously release them all into fusion, we increased the time cells were mated in the dark. However, fusion efficiency dropped when cells were

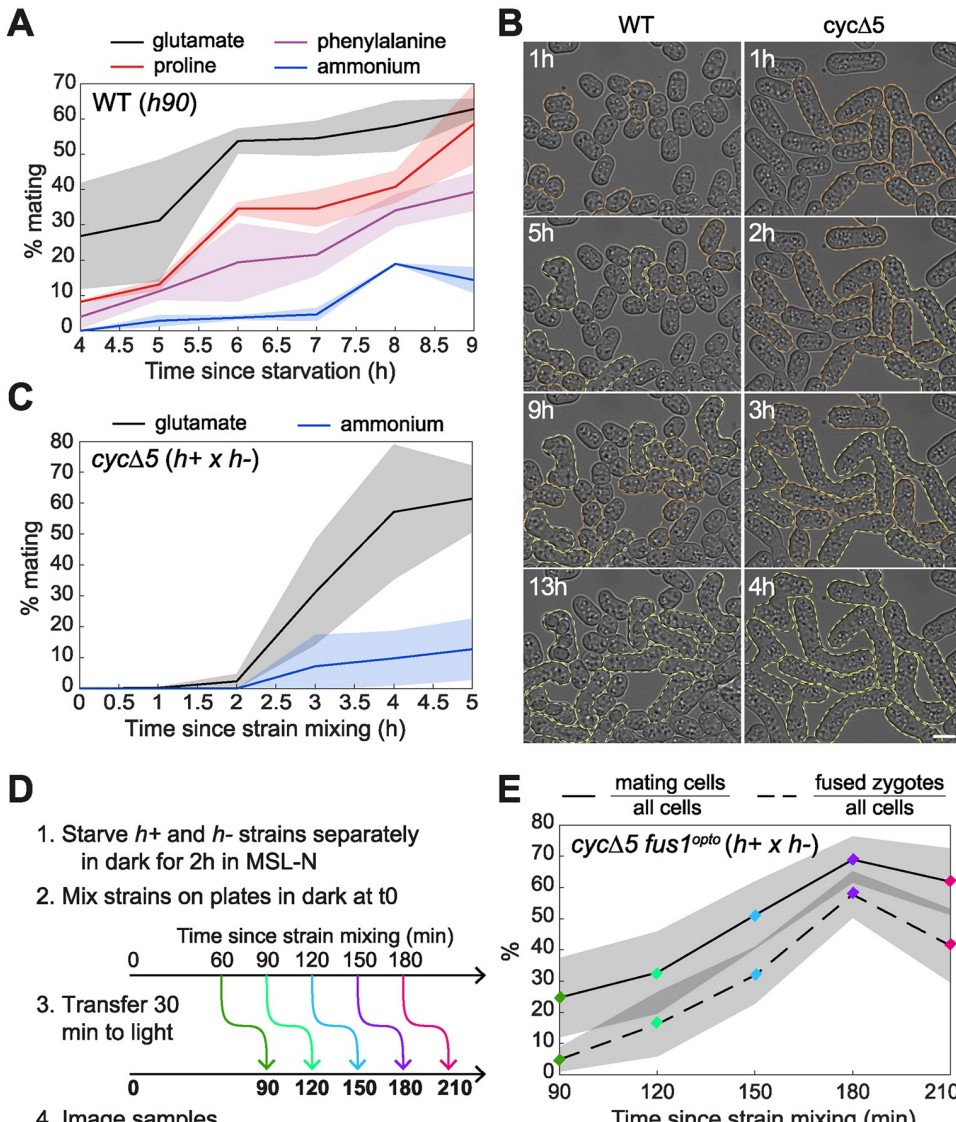

**Fig 2. Optimization of mating and fusion synchrony.** (**A**) Mating efficiency (% cells engaged in mating over total population) of *h90* WT prototroph strain pre-grown in MSL medium supplemented with 15 mM of the indicated nitrogen source and shifted to MSL-N at t = 0. *N* = 2 experiments with ≥228 cells quantified per time point. (**B**) Time-lapse DIC images of *h90* WT (left) and *h+* x *h- cycΔ5* (right) on MSL-N pads. Orange dashed lines highlight mating pairs; yellow dashed lines highlight zygotes. Time is relative to starvation. Note the faster timing of *cycΔ5* cells. Scale bar = 5 μm. Full-size time-lapse available at 10.6084/m9.figshare.27756342. (**C**) Mating efficiency of *h+* and *h- cycΔ5* strains pre-grown in MSL + ammonium or glutamate and mixed in MSL-N at t = 0. *N* = 2 experiments with >350 cells quantified per time point. (**D, E**) Timeline (D) and quantification (E) of the synchronization experiment where *h+* and *h- cycΔ5 fus1^opto* cells were pre-grown in MSL + glutamate, individually starved for 2 h in MSL-N and mixed on MSL-N plates at t = 0. Cells were kept in the dark throughout, and then shifted at indicated time to light for 30 min to activate Fus1^opto before imaging. In (E), mating efficiency and % of fused pairs (dotted line) were quantified over the total cell population. *N* = 4 experiments with ≥1,023 cells quantified per time point. The underlying data for panels A, C, and E can be found in S1 Data.

held in the dark for longer times (Fig 1E), suggesting that cell pairs held in a pre-fusion state eventually lose fusion competence. Thus, the *fus1^opto* allele works well for early mating cells but requires faster sexual differentiation and cell pairing to synchronize the entire cell population.

## Conditions to increase mating speed and synchrony

The G1 phase of the cell cycle is the only permissive phase for sexual differentiation [10,40]. Fission yeast cells grown in presence of ammonium as main nitrogen source exhibit a very short G1 phase and spend most of their cell cycle in G2 phase. Therefore, in the population, only a small fraction of cells is in G1 phase. This provides one explanation for the slow, asynchronous mating process. Indeed, starved WT homothallic cells pre-grown in MSL-ammonium medium slowly accumulate zygotes over >20 h [39]. We reasoned that increasing the fraction of G1 cells in the population would hasten the process. Entry into S-phase and thus the relative length of G1 and G2 phase can be modulated by nitrogen sources [41,42]. We tested glutamate, phenylalanine, and proline as alternatives to ammonium. Pre-growing cells on medium containing these amino acids as main nitrogen source led to faster mating upon nitrogen starvation (Fig 2A). Surprisingly, the fastest mating was observed with pre-growth on glutamate, a source slightly poorer than ammonium [23], but better than proline [42–44]. The change was only in the rate of mating, not on the overall mating or fusion success (S1A and S1B Fig). We thus used glutamate to hasten sexual differentiation in all subsequent experiments.

As a second way to slow down S-phase entry and increase G1 phase, we deleted all non-essential cyclins (*cig1Δ cig2Δ crs1Δ puc1Δ rem1Δ*, here named *cycΔ5*). These cells exhibit a delay in G1/S transition, resulting in an increased cell size (Fig 2B) and meiotic defects [32,45]. However, these cells are highly mating competent, yielding similar pairing and fusion efficiencies as WT cells after 24 h of starvation (S1A and S1B Fig). In addition, they mate substantially faster than WT cells. Like WT cells, *cycΔ5* cells differentiated faster when pre-grown on glutamate than ammonium, with maximal mating reached about 6 h after starvation (*h+* and *h-* strains were pre-starved separately for 2 h before mixing; Fig 2B). In these conditions, mating was remarkably synchronous, going from none to 60% of the population between 2 and 4 h post-mixing (Fig 2C).

Finally, we combined the *cycΔ5* and *fus1^opto^* alleles to create optimal synchronization conditions for mating and cell fusion. We conducted the following experiments (as well as all the phosphoproteomics preparations below) on agar plates, on which we observed even faster and more extensive mating than in liquid cultures, with 70% of cells forming pairs already 180 min after mixing (Fig 2D). We note that in the *cycΔ5* fast mating conditions, the Fus1N-CIBN weak fusogenic activity was not visible, as *cycΔ5 fus1^opto^* cells held in the dark for up to 2 h 30 post-mixing were not observed to form zygotes (S1C Fig). We probed for the optimal time to induce cell fusion by shifting the plate to the light for 30 min after varying amounts of time in the dark. Maximal zygote fraction (57.8% of the entire cell population) was seen after 180 min (150 min dark, 30 min light). In summary, we set up conditions for fast sexual differentiation and synchronous cell fusion in *cycΔ5 fus1^opto^* cells, which allows to perform population-based time course studies.

## Phosphoproteomic time-course analyses of nitrogen starvation

To describe the phosphorylation changes occurring during sexual differentiation, we collected protein extracts at 45 min interval of *h-* and *h+* cells, mixed on agar plates lacking nitrogen, and kept in the dark to block cell–cell fusion (mating samples) (Fig 3A). To discriminate the changes induced by pheromone signaling from those induced by starvation and transfer to solid medium, we also collected extracts of *h-* and *h+* cells plated on distinct petri dishes and mixed only at the time of extract preparation (starvation samples). Using 10plex tandem mass tag (TMT), 3 biological replicates were then analyzed by proteomics and, after phospho-peptide enrichment, by phosphoproteomics. A total of 2,851 proteins (out of 5,133 coding genes

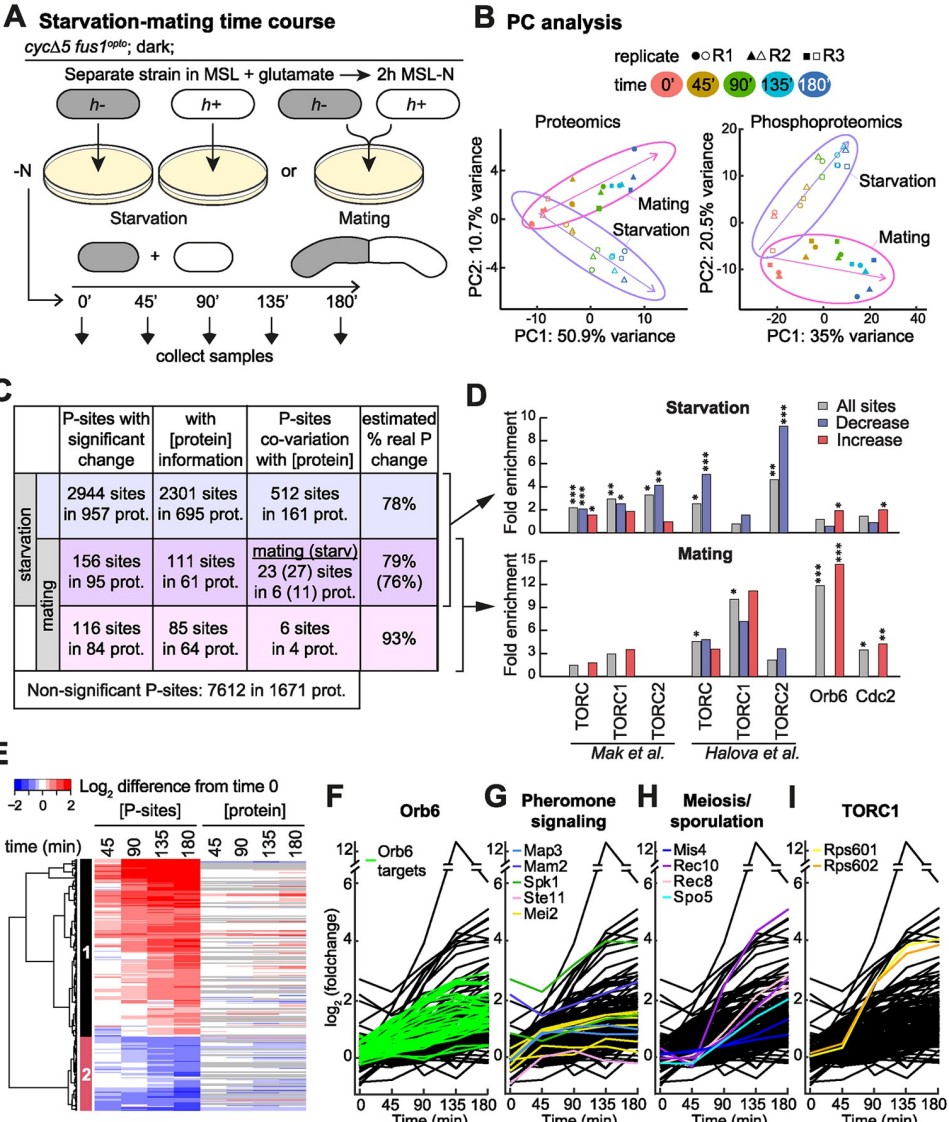

**Fig 3. Proteomic and phosphoproteomic time course over sexual differentiation.** (**A**) Experimental design. *h+* and *h- cycΔ5 fus1^opto^* cells were either plated at a 1:1 ratio on the same MSL-N plate (mating samples) or onto separate MSL-N plates (starvation samples) and mixed only at the time of sample collection, every 45 min from time of plating. (**B**) PC analysis on median-normalized proteomic (left) and phosphoproteomic (right) data. Purple and magenta arrows represent the direction taken by the mating and starvation samples, respectively, in the PC space. Note that the first PC separates time and the second PC distinguishes starvation and mating samples. (**C**) Table displaying the number of phosphosites with significant changes during starvation (top 2 lines) and with significant difference between mating and starvation conditions (bottom 2 lines); 272 (156 + 116) sites changed specifically during mating, after correction by the starvation values. Where protein data was available (column 2), linear regression was used to probe phosphosite co-variation with protein abundance (column 3), showing that changes in the majority of phosphosites occur independently of protein abundance (right column). For phosphosites changing significantly upon starvation and differently during mating (middle line), co-variation was analyzed with protein levels in both starvation and mating samples. Details of all sites can be found in S1 and S2 Tables and on https://pombephosphoproteomics.unige.ch/. (**D**) Fold enrichment for TORC, TORC1, TORC2, Orb6, and Cdc2 substrates among phosphosites that change (gray), decrease (blue) or increase (red) in abundance during starvation (top) and mating (bottom). Fisher-exact tests were used. *: $p < 0.05$, **: $p < 0.01$, ***: $p < 0.001$. The underlying data can be found in S1 Data. (**E**) Heatmap of the starvation-corrected values for the 272 phosphosites that change specifically during mating. Protein dynamics shown on right with missing values in gray. The underlying data can be found in S1 Table. (**F–I**) Graph representation of the phosphosite values increasing during mating with colored traces for Orb6 target sites (F), proteins involved in pheromone signaling (G), proteins involved in meiosis or sporulation (H), and ribosomal protein S6 (I). The underlying data can be found in S1 Table; Orb6 targets are indicated in S1 Data.

in the *S. pombe* genome) and 10,828 phosphosites in 1,884 proteins were identified in at least 2 biological replicates. After data normalization to correct for batch (replicate) effect, principal component (PC) analysis revealed good clustering of replicates and stratification of starvation and mating data for both proteomic and phosphoproteomic data, with PC1 reflecting time and PC2 reflecting major differences between starvation and mating (Fig 3B). All proteomic and phosphoproteomic changes described in this study are described in S1 and S2 Tables, as well as on an associated website for easy browsing (https://pombephosphoproteomics.unige.ch/). Phosphorylation changes were highly reproducible across biological replicates (see website and S2A Fig for some examples).

Even though the starvation data set does not probe for acute changes in the (phospho)proteome upon starvation, as strains were pre-starved in liquid conditions for 2 h before plating, control samples showed massive changes in the proteome and phospho-proteome, with at least 847 proteins and 3'100 phosphosites in about 1'000 proteins found to exhibit significant change in amounts over the 3 h time course (S3A and S4A Figs and S1 Table). Because of the massive change in protein amounts, one concern in analyzing the phosphoproteome is that changes in phosphosite abundance may be a consequence of change in protein levels rather than a regulatory change. To probe for this possible confounding factor, we compared the slope of the changes in protein and phosphosite amounts over time. This analysis revealed that at least 78% of sites vary independently of protein concentration (Fig 3C). Thus, even if a substantial fraction of phosphosites is concordant with protein abundance, global phosphoproteomic changes are not dominated by changes in protein levels (S4A Fig).

Gene ontology (GO) analysis showed that proteins involved in catabolic processes were enriched in the group whose levels increased, whereas proteins involved in biosynthetic processes, such as amino-acid biosynthesis, ribosomal, and translation functions, were enriched in the group whose levels decreased during starvation (S3B and S3C Fig). This is in agreement with a general switch-off of de novo protein translation and the known re-use of existing cellular material, in particular through autophagy, upon nitrogen depletion [46]. We indeed observed an enrichment in proteins involved in autophagy in the group whose phosphorylation increases during starvation (S4B Fig). Interestingly, mitochondrial respiration functions also featured prominently in proteins increasing during starvation, perhaps reflecting the recently established link between starvation-induced autophagy and respiration [47] (S3B Fig). Significant changes in phosphorylation patterns included GO terms in autophagy, transmembrane transport, and cell cycle regulation, but also affected different classes of functions, mostly relating to the cytoskeleton and cell polarity (S4B and S4C Fig). Comparison with prior phosphoproteomic studies revealed a small enrichment in targets of Orb6 and CDK1 kinases in phosphosites increasing during starvation [34,48] (Fig 3D, top). Strikingly, phosphosites decreasing during starvation were highly enriched in candidate TOR targets identified in 2 previous phosphoproteomic studies [49,50]. While the attribution of sites to TORC1 or TORC2 has been difficult due to absence of specific TORC2 inhibitors, and differs in the 2 studies, a loss of TORC1-dependent phosphorylation is expected from the well-established inactivation of TORC1 upon starvation.

Overall, our global analysis of proteomic and phosphoproteomic changes upon nitrogen starvation indicates a considerable re-wiring of cellular metabolism, reducing de novo protein translation and increasing nitrogen and carbon catabolism, likely imposed in part by reduction in TORC1 activity. While this forms a rich data set to explore the signaling consequences of nitrogen starvation, we did not investigate it further in this study. Instead, we used it as normalizing information to identify phosphosites changing specifically during mating.

## Phosphoproteomic time-course analyses of sexual differentiation

To identify changes specific to mating, we looked for significant changes after subtraction of the starvation signal from the mating one. Specifically, we fit a linear model with the values $(i_t^{mating} - i_{t-1}^{mating}) - (i_t^{starvation} - i_{t-1}^{starvation})$, where $i_t^{mating}$ is the intensity of a given peptide in the mating sample at time t. Only 54 proteins showed significant changes in abundance (adjusted p-value <0.05; S5A Fig), most of which increased. Consistent with the known positive feedback between pheromone signaling and transcriptional control during sexual differentiation [26], proteins with increased abundance were enriched in pheromone signaling proteins, but also in cell wall remodeling factors (S5B Fig). We also identified a total of 272 phosphosites with significant changes along the mating time course. Of these, 116 changed only during mating and 156 also significantly changed upon starvation but changed differently in the presence of mating partners (Fig 3C and 3E; see also S2A Fig for examples of phosphorylation profiles during starvation, mating, and the corrected mating-starvation signal). Where proteomic information was available, we explicitly investigated which of these variations were independent of protein abundance, which showed that 79% to 93% of phosphorylation changes are independent of protein level changes (Fig 3C and 3E). Thus, compared to nitrogen starvation, the presence of a mate induces relatively modest changes in the proteome and phospho-proteome, which are largely independent of each other.

The phosphosites can be grouped in 2 clusters whose intensity increases or decreases monotonically during sexual differentiation (Fig 3E). GO term enrichments were highly divergent in the 2 groups (S5C and S5D Fig). Proteins involved in G1/S transition, including the MBF transcription complex (Cdc10, Res1, and Whi5), actin cytoskeleton organization, vesicle-mediated transport, cell polarity or localization to the cell tip were highly enriched in the group showing increased phosphorylation. Proteins involved in amino acid, peptide, and nucleobase transport were strongly enriched in the dephosphorylated group. In both groups, protein localization to the plasma membrane and the Golgi were overrepresented. These phosphorylation changes at the cell periphery are consistent with G1 arrest and reorganization of the polarity and actin machinery for shmoo growth and in preparation for cell–cell fusion. Examination of the sequence around phosphorylation sites showed overrepresentation of proline after the phospho-S/T residue, a motif typical of CDK and MAPK, and overrepresentation of basic residues before the phospho-S/T, characteristic of the group of AGC kinases (S5E Fig, left) [51,52]. The substrates of the pheromone-MAPK Spk1 await identification, but consistent with these observations, comparison with available phosphoproteome data sets showed a small enrichment in targets of CDK1 and very large enrichment in targets of the AGC group NDR/LATS-family Orb6 kinase (Fig 3D, bottom and Fig 3F). Orb6 is essential for mitotic growth and viability, and its activity is down-regulated upon nitrogen starvation [53,54]. This suggests that Orb6 is re-activated upon pheromone signaling when its functions in cell polarization and vesicular trafficking described during mitotic growth are likely repurposed for shmoo growth. The enrichment for TOR targets associated with dephosphorylated peptides during nitrogen starvation was largely lost in our mating-specific data set, which discounts starvation signals, although surprisingly some enrichment remained significant with one of the 2 TOR target data sets (see below). We note that the overrepresentation of basic residues before the phospho-S/T remained after removal of all known Orb6 targets, suggesting activation of additional basic residues-directed kinases (S5E Fig, right).

Among sites whose phosphorylation increases, we identified several expected sites on proteins necessary for mating (colored in Figs 3G and S2B), including in the pheromone MAPK-Spk1 active site (T199 and Y201) [55] and on the C-terminal tails of pheromone receptors, likely to promote their internalization (T325 on Mam2 and S346, S349 on Map3) [56].

Phosphorylation of the transcription factor Ste11 also significantly increased after 45 min, though surprisingly not at the previously mapped MAPK-Spk1 target sites [57], but at T173 shown to be phosphorylated by Pat1 kinase [58]. Increase in Mei2 phosphorylation was also detected on 6 sites (S60, S171, S173, S420, S454, and T573), all different from the inhibitory Pat1 and TORC1 target sites [59,60], suggesting Mei2 is not only de-phosphorylated for mating induction but that other phosphorylation events also promote its activity.

Unexpectedly, several proteins with specific function in meiosis showed significant increase in phosphorylation during mating (pre-fusion) (Fig 3H). These phosphorylation events are not due to leakage fusion, as cells held in the dark showed no fusion in the phosphoproteomic screen conditions (S1C Fig). These include Num1/Mcp5, the meiotic cortical dynein anchor, necessary for horsetail movement [61,62]; Rec10, a component of the meiotic linear element, functionally similar to the synaptonemal complex, on sites S347 and S529 previously identified in diploid cells [63], though with unclear functional significance [64]; the meiotic cohesin sub-unit Rec8 on sites S366 and S434, the first of which is phosphorylated by Casein kinase 1 to promote cleavage by separase [65]; the cohesin loading factor Mis4; and the RNA-binding protein Spo5, previously thought to be specifically expressed during meiosis [66]. Spo5 has functions in sporulation and in promoting cyclin Cdc13 expression for meiosis II [67], and we found phosphorylation increase on sites necessary for timely degradation at meiosis II [68]. Thus, expression and phosphorylation of (at least some) zygotic meiotic and sporulation factors occurs ahead of gamete fusion.

Finally, we observed an unexpectedly large phosphorylation increase in Rps6 S235-S236 phosphorylation (Fig 3I; see below).

## Phosphoproteomic time-course analysis of cell–cell fusion

To probe for specific phosphorylation changes over the fusion process, *h-* and *h+* cells were mixed on agar plates lacking nitrogen and kept in the dark for 150 min. Protein extracts were collected upon illumination at 11 min interval (fusion samples; Fig 4A). Because *S. pombe* does not encode light-sensing proteins, we did not specifically control for the light conditions. Data acquisition and analysis was performed as described above, but with 4 biological replicates. PC analysis showed decent clustering of replicate time points, with an interesting trajectory showing an inflexion at 22 min (Fig 4B), corresponding to the median fusion time (see Fig 1C). We identified 41 significant proteomic (S6A and S6B Fig) and 440 significant phosphoproteomic changes, 55 of which overlapped with sites shown to vary in the mating time course (Fig 4C). As in the mating time course, most phosphorylation changes were independent of protein intensity changes (Fig 4C and 4E).

These phosphosites were grouped in 3 clusters: profiles in cluster 1 increased already over the first 11 to 22 min, those in cluster 2 increased only between 22 and 44 min, and those in cluster 3 decreased over the time course (Fig 4E–4H). Because 20 min is the median time from illumination to cell fusion (see Fig 1C), the 2 classes of phosphorylation increase suggest that some sites are phosphorylated for cell fusion, whereas others are phosphorylated in the zygote in response to cell fusion perhaps to help terminate the process and prepare for meiosis and sporulation. Cluster 2 includes for instance Rep1/Rec16, which both promotes early meiotic gene expression and represses pheromone signaling genes [69], Spo5, Mis4, Rec10, and Rec8, which is further phosphorylated on multiple Casein kinase I and Polo kinase target sites [65,70,71] (Fig 4G). Both increasing clusters showed similar overrepresentation of actin-associated and cell tip-localized proteins (S6C Fig). Enrichment for substrates of known kinases suggested a possible enrichment in TORC2 targets among phosphosites that increased in abundance, which may be consistent with its role in regulating the actin cytoskeleton (Fig 4D).

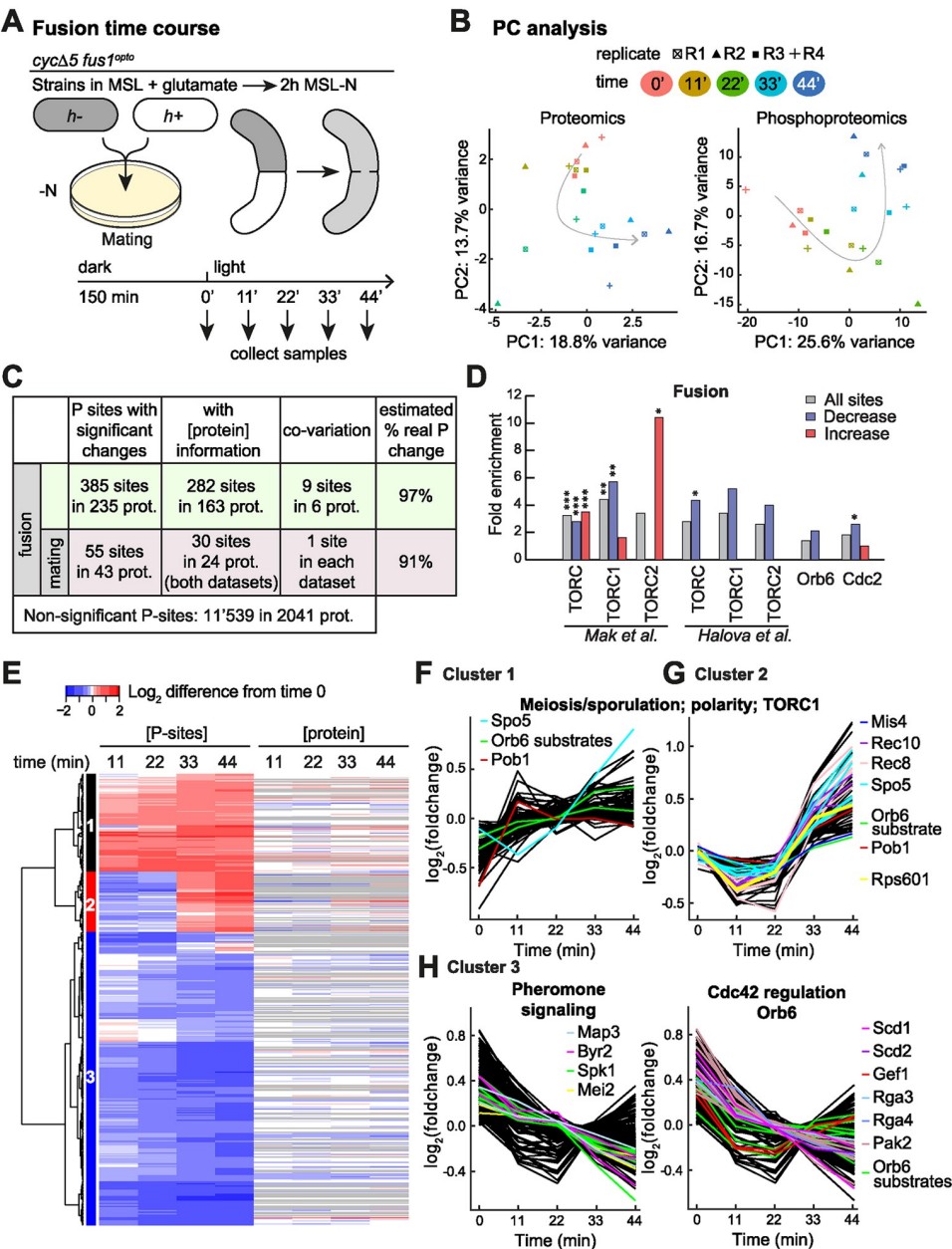

**Fig 4. Proteomic and phosphoproteomic time course over cell–cell fusion.** (**A**) Experimental design. *h+* and *h-cycΔ5 fus1^opto* cells were plated at a 1:1 ratio on MSL-N plates (as for mating samples) incubated in the dark for 135 min and then shifted to light and samples collected every 11 min. (**B**) PC analysis on median-normalized proteomic (left) and phosphoproteomic (right) data. Arrows represent the direction over time in the PC space. Note the inflexion at 22 min, corresponding to median fusion time (see Fig 1C). (**C**) Table displaying the 440 (385 + 55) phosphosites with significant changes during cell fusion. A total of 55 sites were also changing significantly in the mating samples (bottom line). Where data was available (column 2), linear regression was used to probe phosphosite co-variation with protein abundance (column 3), showing that changes in the majority of phosphosites occur independently of protein abundance (right column). Details of all sites can be found in S1 and S2 Tables and on https://pombephosphoproteomics.unige.ch/. (**D**) Fold enrichment for TORC, TORC1, TORC2, Orb6, and Cdc2 substrates among phosphosites that change (gray), decrease (blue) or increase (red) in abundance during cell–cell fusion. Fisher-exact tests were used. *: $p < 0.05$, **: $p < 0.01$, ***: $p < 0.001$. The underlying data can be found in S1 Data. (**E**) Heatmap of the values for the 440 phosphosites that changed during cell fusion. Protein dynamics shown on right with missing values in gray. The underlying data can be found in S1 Table. (**F–H**) Graph representation of the phosphosite values as in (E) for phosphosites showing increase pre-fusion (F), post-fusion (G), and decrease (H) with example proteins showed with colored traces. The underlying data can be found in S1 Table; Orb6 targets are indicated in S1 Data.

Among protein functions showing phosphorylation decrease, there was a strong overrepresentation of pheromone-MAPK cascade and polarity factors (Figs 4H and S6D), suggesting that some factors phosphorylated for cell pair formation during mating become dephosphorylated during fusion. These include, for pheromone signaling, the MAPK Spk1, the pheromone receptor Map3, on the same sites that showed increase pre-fusion and the MAPKKK Byr2 (Fig 4H, left). For polarity, we found many Orb6 targets as well as regulators and effectors of the small GTPase Cdc42 (the GEFs Scd1 and Gef1, scaffold Scd2, 2 Cdc42 GAPs and effector Shk2/Pak2 kinase) (Fig 4H, right).

Finally, we again identified an increase in Rps6 S235-S236 phosphorylation (Fig 4G).

## Reactivation of TORC1 by the pheromone-MAPK pathway during nitrogen starvation

Our mating and fusion phosphoproteomic data sets revealed strong phosphorylation of ribosomal protein S6 (Rps6), which is encoded by 2 paralogous genes (*rps601* and *rps602*), on the conserved residues S235-S236 (Figs 3I and 4G). This phosphorylation event, which is conserved across eukaryotes, primarily depends on S6 kinases (S6K) downstream of TORC1 and is widely used as a marker of TORC1 activity [72]. In fission yeast, Rps6 is phosphorylated during mitotic growth by the S6K-like kinase Psk1 downstream of TORC1 [73,74]. However, upon nitrogen starvation, TORC1 activity is down-regulated, and down-regulation of TORC1 activity is necessary and sufficient for the initiation of sexual differentiation [11,12]. Rps6 is also phosphorylated by the AGC family kinase Gad8 downstream of TORC2 [75], which positively regulates mating [21,76,77]. We thus tested which TOR complex controls Rps6 phosphorylation during mating.

We first verified the phosphoproteomic data by using phospho-specific antibodies against Rps6 S235-S236 phosphorylation in both *cyc5Δ* and WT cells. As expected, heterothallic *h-* strains showed low (but non-zero) Rps6 phosphorylation on S235-S236 after 6 h 30 (3 h for the *cyc5Δ* background) of nitrogen starvation. By contrast, homothallic *h90* WT strains (which contain a mix of the 2 cell types and thus undergo sexual differentiation) or a mix of *h+* and *h-cycΔ5* cells showed high Rps6 phosphorylation at the same time points (Fig 5A), confirming that Rps6 is phosphorylated during sexual differentiation also in WT cells. To probe which of the 2 TOR complexes functions upstream of Rps6 phosphorylation, we treated cells with Torin1, which inhibits both TOR complexes, and rapamycin, which specifically inhibits TORC1 [73,78]. Drugs were added for 30 min on already differentiated cells. Rps6 phosphorylation was reduced in both conditions (Fig 5B), suggesting dependence on TORC1 activity. Rps6 global levels were unchanged in these experiments.

Because rapamycin has additional effects, including on Tor1, the main TORC2 kinase [21,79], and on the FKBP12 homologue Fkh1 during mating [80], we sought additional evidence to support the requirement for TORC1 activity. First, we found that the S6K-like Psk1 kinase, a direct substrate of TORC1 which in turn phosphorylates Rps6 during mitotic growth [74], was strongly phosphorylated on T415 in *h90* but not in *h-* strains upon N-starvation, and that this phosphorylation was rapamycin dependent (Fig 5B). 5 μm Torin1 was inefficient in the experiment shown in Fig 5B, but we confirmed that 25 μm Torin1 inhibited both Rps6 and Psk1 phosphorylation (S7A Fig). By contrast, phosphorylation of the main TORC2 substrate Gad8 was largely unchanged in *h90* strains and upon rapamycin treatment (Fig 5B), confirming that rapamycin does not affect TORC2 activity. We note that Gad8 was destabilized upon Torin1 treatment (Fig 5B). Second, the rapamycin-insensitive *tor2* allele, *tor2-S1837E*, prevented loss of Rps6 and Psk1 phosphorylation upon rapamycin treatment, demonstrating that rapamycin inhibits these phosphorylation events by binding to the TORC1 kinase Tor2 (Fig 5C).

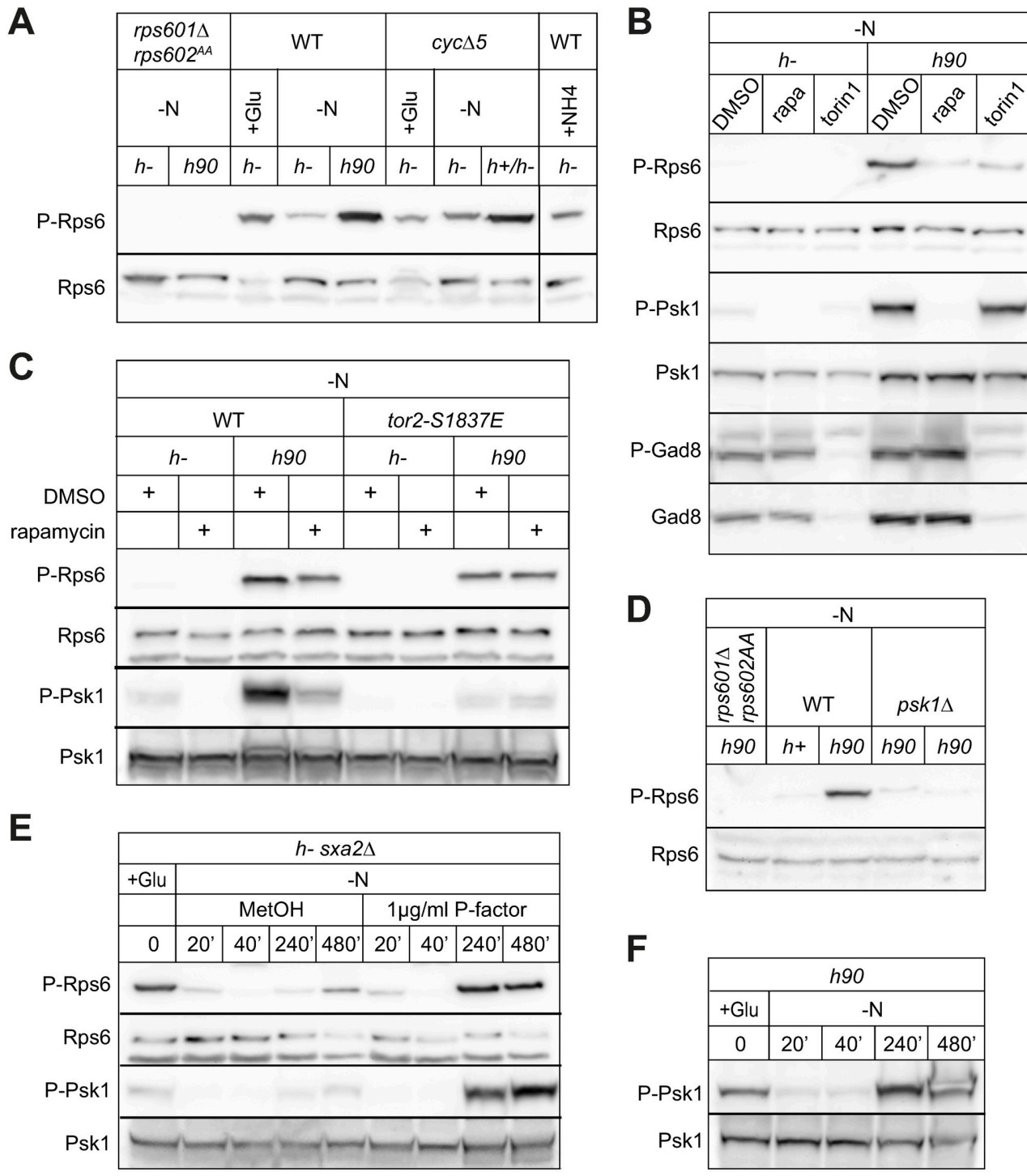

**Fig 5. TORC1 re-activation by pheromone signaling.** (**A**) Western blot of Rps6 phosphorylation in WT and *cycΔ5 fus1^opto* cells kept in the dark. Cells were either grown in MSL + ammonium sulfate (+ NH4), MSL + Glutamate (+Glu) or starved for 6 h 30 (WT) or 3 h (*cycΔ5 fus1^opto*) without nitrogen (-N). Homothallic strains (*h90*) or mix of mating types (*h+/h-*) mate in these conditions. The *rps601Δ rps602^AA* strain was used as control for the specificity of the phospho-specific antibody. Rps6 total levels are shown at the bottom. (**B**) Phospho-Rps6, phospho-Psk1, and phospho-Gad8 levels in WT heterothallic (*h-*) and homothallic (*h90*) cells starved for 6 h 15 and then treated with either rapamycin (300 ng/ml), Torin1 (5 μm), or DMSO for 30 min. See S7A Fig for further Torin1 treatment. (**C**) Phospho-Rps6 and phospho-Psk1 in WT and *tor2^S1837E* mutant treated with rapamycin or DMSO as in (**B**). (**D**) Phospho-Rps6 levels in wild type and *psk1Δ* mutant after 6 h 30 starvation. (**E**) Time course of phospho-Rps6 and phospho-Psk1 in *h- sxa2Δ* cells starved in the presence of P-factor (1 μg/ml) or MetOH. Starvation and P-factor addition occurred at T = 0 min. (**F**) Time course of phospho-Psk1 in h90 WT cells starved at T = 0 min. Uncropped western blots available in S1 Raw Images.

Interestingly, *tor2-S1837E* was also hypomorphic for Psk1 phosphorylation (Fig 5C). We further confirmed that Psk1 phosphorylates Rps6 also during mating (Fig 5D). We conclude that phosphorylation of Rps6 during mating depends on TORC1 signaling through Psk1 kinase.

Sexual differentiation requires pheromone signaling through a GPCR-Ras-MAPK cascade functionally similar to the ERK cascade in mammalian cells. The cascade can be elicited in heterothallic cells by exposure to synthetic pheromone normally produced by partner cells. Nitrogen starvation of *h-* cells (lacking the P-factor protease Sxa2) led to acute loss of Psk1 and Rps6 phosphorylation over the first 20 to 40 min of starvation, followed by signal re-appearance after 4 to 8 h (Fig 5E). This re-appearance of phospho-Rps6 and phospho-Psk1 after a few hours of starvation reflects the documented autophagy-mediated reactivation of TORC1, likely in response to autophagy-dependent release of intracellular nutrients [81–83]. Addition of 1 μg/ml P-factor did not change the acute loss of phosphorylation upon starvation, but strongly boosted their re-appearance (Fig 5E). We also observed similar dynamics when probing for phospho-Psk1 in a time course of mating cells (Fig 5F). Thus, pheromone signaling promotes TORC1 re-activation.

## Pheromone-dependent reactivation of TORC1 does not require autophagy

We probed for possible links between pheromone-dependent TORC1 re-activation and autophagy. In nitrogen-free medium, autophagy-deficient mutants are unable to initiate sexual reproduction due to their inability to recycle proteins to basic building blocks [84,85]. However, they are fertile if induced to mate in low levels of nitrogen [86]. We established that a shift from MSL + 15 mM to MSL + 0.5 (or 0.75) mM glutamate allowed both *h- sxa2Δ* (WT) and *h- sxa2Δ atg1Δ* mutants to form shmoos in response to 1 μg/ml P-factor (Fig 6A). We first used these conditions to probe whether starvation-dependent induction of autophagy is modulated by pheromone signaling. As a readout, we used CFP-Atg8 cleavage, as CFP-Atg8 is delivered to vacuoles by the autophagic process, where only the CFP moiety is resistant to vacuolar proteases. In WT cells, CFP-Atg8 cleavage occurred upon transfer to low-glutamate conditions and increased over time in a similar manner whether cells were treated with MetOH or P-factor (Fig 6B). This suggests that pheromone signaling does not induce autophagy.

When we probed CFP-Atg8 cleavage in *atg1Δ* as control, we were surprised to observe significant residual levels of free CFP (Figs 6C and S7B), as deletion of this upstream regulatory kinase is reported to fully block autophagy [85,87,88]. We reasoned that this difference may be due to differences in media or nitrogen source, as we used MSL + glutamate, while earlier studies used EMM + ammonium. Indeed, whereas exchanging glutamate for ammonium during growth before starvation had no effect on the observed cleavage in MSL medium (S7B Fig), using EMM medium abrogated cleavage (Fig 6C), as previously reported. Thus, for currently unknown reasons, deletion of Atg1 kinase only partially prevents autophagy in MSL medium.

To identify autophagy-deficient mutants in MSL medium, we screened through several *atg* mutants for those that would block CFP-Atg8 cleavage. This revealed that *atg* mutants exhibit a range of residual cleavage after 4 h starvation in MSL-N, with deletion of *atg5*, *atg7*, and *atg18a* being the most potent in blocking CFP release in these conditions (Fig 6D). We thus chose, in addition to *atg1Δ*, *atg5Δ*, and *atg18aΔ*, which also responded to P-factor in our established conditions (Fig 6E), to test for Psk1 phosphorylation. In WT (*sxa2Δ*), Psk1 phosphorylation loss was less acute upon transfer to low-glutamate than nitrogen-free medium, as expected. At late time points, Psk1 was further de-phosphorylated in DMSO-treated samples and was strongly re-phosphorylated in response to P-factor, as observed upon complete nitrogen starvation (Fig 6F). In *atg1Δ*, *atg5Δ*, and *atg18aΔ* mutants, the dynamics of Psk1

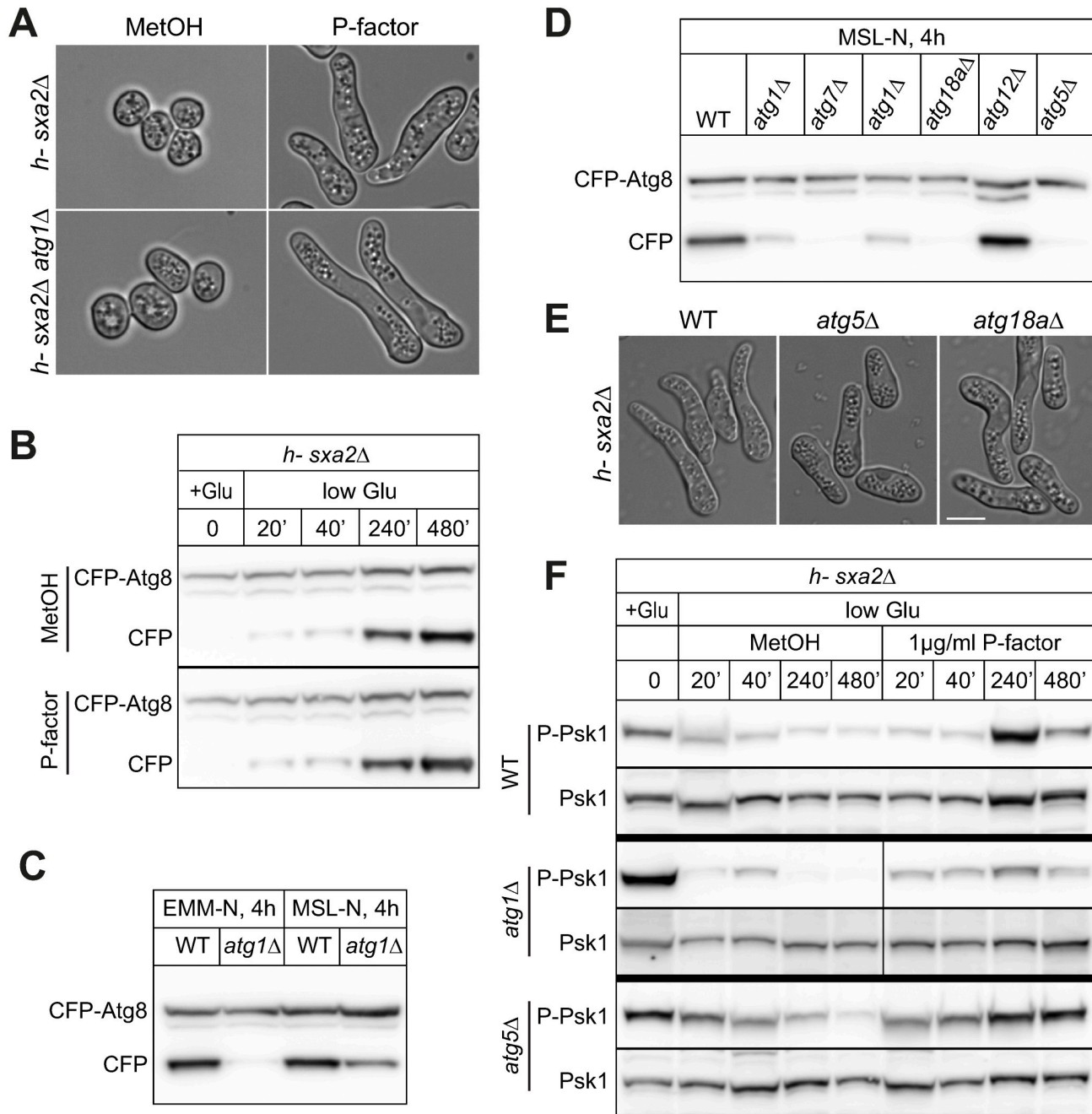

**Fig 6. Pheromone-dependent TORC1 re-activation is independent of autophagy.** (**A**) *h- sxa2Δ* cells, which are otherwise WT (top) or *atg1Δ* (bottom), after 1 μg/ml P-factor or MetOH treatment for 24 h in MSL + 0.5 mg/ml glutamate. P-factor induces shmoo formation in both strains. Original data available at 10.6084/m9.figshare.27756342. (**B**) Time course of CFP-Atg8 cleavage in *h- sxa2Δ* cells transferred to MSL + 0.5 mg/ml glutamate in the presence of P-factor (1 μg/ml; bottom) or MetOH (top) at T = 0 min. (**C**) *atg1Δ* cells are not fully autophagy-deficient in MSL medium. CFP-Atg8 cleavage in WT or *atg1Δ* cells after 4 h of nitrogen starvation in EMM or MSL. These strains are also *sxa2Δ* as in (A) and (F). (**D**) CFP-Atg8 cleavage in indicated mutants after 4 h of nitrogen starvation in MSL. The strains in the first 2 lanes are also *sxa2Δ* as in (A) and (F). (**E**) *h- sxa2Δ* cells with indicated additional deletion after 1 μg/ml P-factor treatment for 24 h in MSL + 0.75 mg/ml glutamate. Shmoo form in all strains. Original data available at 10.6084/m9.figshare.27756342. (**F**) Time course of phospho-Psk1 in *h- sxa2Δ* cells, which are otherwise WT (top), *atg1Δ* (middle), or *atg5Δ* (bottom), transferred to MSL + 0.75 mg/ml (or 0.5 mg/ml in case of *atg1Δ*) glutamate in the presence of P-factor (1 μg/ml) or MetOH at T = 0 min. Please see S7D Fig for CFP-Atg8 in these same extracts. Uncropped western blots available in S1 Raw Images.

phosphorylation was very similar: in all cases, the P-Psk1 signal initially dropped upon transfer to low glutamate (see 20' and 40' time points). In control conditions (MetOH), it then stayed low or was even further reduced at late time points (240' and 480'). By contrast, in presence of P-factor, Psk1 was re-phosphorylated at these late time points, as happens in WT cells (Figs 6F and S7C; compare the 240' and 480' time points for P-factor versus MetOH control). Western blots of CFP-Atg8 cleavage over the time course confirmed strong reduction of autophagy in the *atg5Δ* and *atg18aΔ* mutants (S7C and S7D Fig). We conclude that pheromone-dependent TORC1 re-activation does not require the autophagy pathway.

## TORC1 activity plays a positive role in sexual reproduction

We probed the possible function of Rps6 and its phosphorylation during sexual reproduction. Previous work had shown that *rps601Δ* and *rps602Δ* are synthetic lethal, but that neither gene alone, nor their phosphorylation, nor the Psk1 kinase, is essential for growth [73]. We found that *psk1Δ* cells are fully competent for mating, and that non-phosphorylatable Rps6 (*rps601Δ rps602^{SSAA}*) is no less competent for mating than the control *rps601Δ* strain, indicating that Psk1 and Rps6 phosphorylation are not critical TORC1 substrates during mating, but merely act as good markers of its re-activation (Fig 7A). Although *rps601Δ* cells had low mating efficiency, we noticed that *rps601Δ* and *rps602Δ* strains were fully sterile in strains auxotrophic for leucine (Fig 7A). While the reasons for this synthetic effect are currently unclear, leucine is a potent direct activator of TORC1 activity in budding yeast and mammalian cells [89,90]. *S. pombe leu1-* cells are rapamycin-sensitive [79] and show fast TORC1 inactivation upon starvation [91], suggesting these cells also have reduced TORC1 activity [92]. We indeed found Psk1 and Rps6 phosphorylation to be reduced during mating in *leu1-32* compared to WT cells, indicating lower TORC1 re-activation (Fig 7B). We had previously noted that *leu1-32* cells exhibit fusion delays and defects [37]. These cells also show strongly reduced mating efficiency (Fig 7A–7C). These correlative data suggest that TORC1 re-activation is required to promote efficient mating.

To probe this point more stringently, we aimed to directly inhibit TORC1 during mating. TORC1 inhibition by rapamycin involves the formation of a trimeric complex between rapamycin, the TOR kinase and the FKBP12 prolyl isomerase, Fkh1 in *S. pombe*. Previous work showed that rapamycin inhibits mating in *S. pombe* [93], but the sterility of *fkh1Δ* cells makes it impossible to interpret whether TORC1 inactivation contributes to this inhibition [80]. TORC1 can also be inhibited by caffeine [94,95] or auxin [96]. Addition of these compounds to WT mating cells led to dose-dependent inhibition of mating, consistent with TORC1 activity acting positively during sexual reproduction (Fig 7D).

We showed above that the rapamycin-insensitive allele *tor2-S1837E* is hypomorphic, as Psk1 phosphorylation is reduced in this strain. In contrast to *tor2* temperature-sensitive mutants, which undergo mating in rich conditions at restrictive temperature [11,12,14], *tor2-S1837E* mutants do not spontaneously mate. By contrast, *tor2-S1837E* mutants were slow to agglutinate in liquid mating reactions (Fig 7E) and produced aberrant tetrads with many inviable spores (Fig 7F). Similarly, tetrads produced by the *tor2-ts10* allele shifted at 32°C to inactivate it only during sexual reproduction were aberrant and contained inviable spores, even if these spores were germinated at permissive temperature (Fig 7F and 7G). Finally, we found that deletion of the non-essential TORC1 component *tco89*, which exhibits reduced TORC1 activity for lifespan regulation and resistance to Torin1 and caffeine [95,97], caused a reduction in mating efficiency (Fig 7H). We conclude that TORC1 re-activation by pheromone signaling is required for correct progression of sexual reproduction.

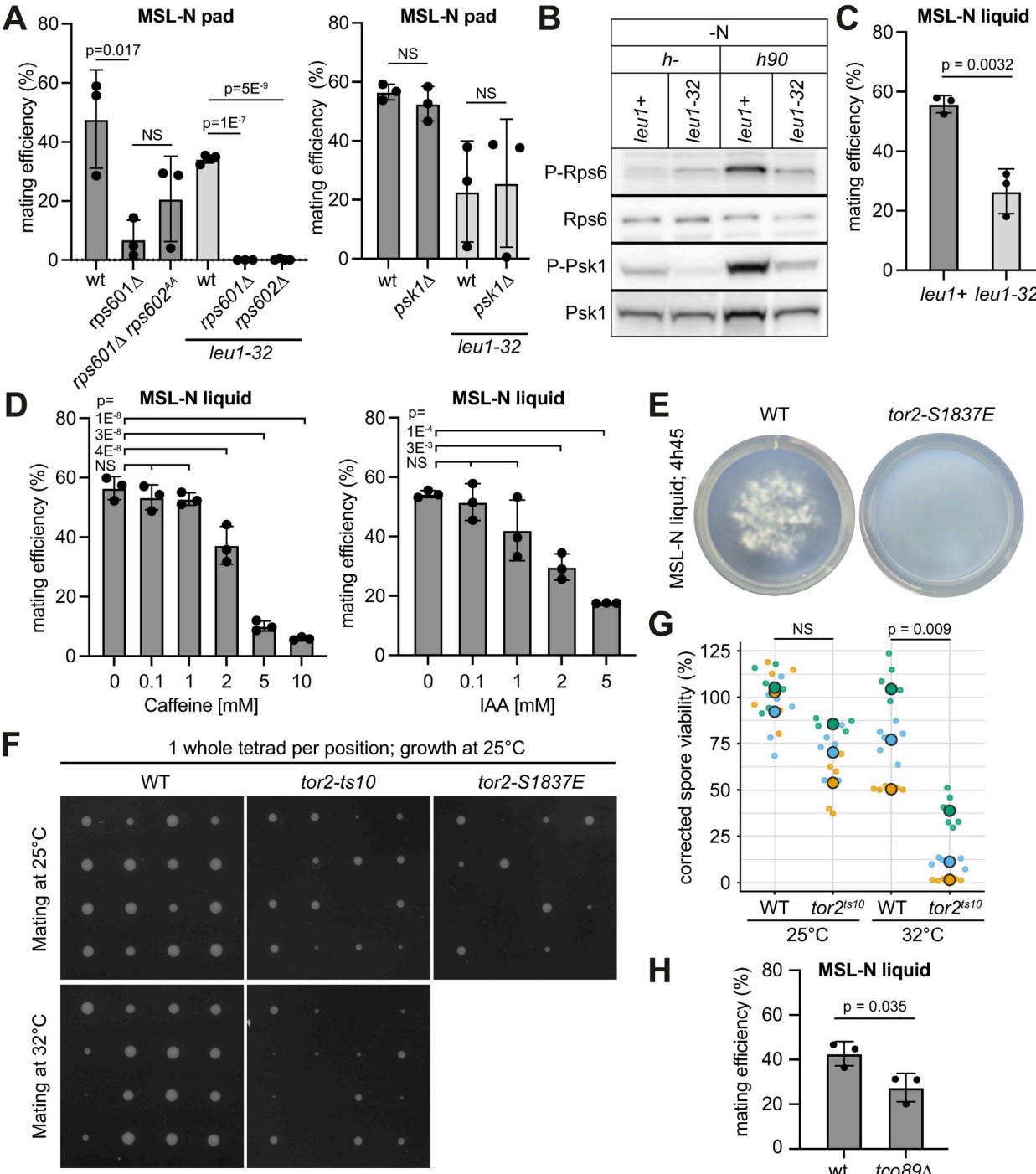

**Fig 7. TORC1 signaling promotes sexual reproduction.** (**A**) Mating efficiency of indicated *h90* strains with or without *leu1-32* auxotrophy. (**B**) Phospho-Rps6 and phospho-Psk1 in starved WT (*leu1+*) and *leu1-32* mutant after 6 h 30 starvation. (**C**) Mating efficiency of *h90* WT (*leu1+*) and *leu1-32* cells in liquid cultures. (**D**) Mating efficiency of WT cells treated with caffeine or auxin (IAA). (**E**) Image of liquid culture of WT and *tor2^{S1839E}* mutants showing cell aggregation of only WT cells. (**F–G**) Spore germination efficiency from tetrads of indicated *h90* strains. (F) Each colony represents the growth from the 4 spores of the same tetrad. (G) % of germinated spores in a random spore analysis. Data are colored by experimental replicate. Dots indicate individual data points (technical replicates). Large circles show averages per replicate. (**H**) Mating efficiency of *h90* WT and *tco89Δ* cells. *T* test *p*-values are indicated except in (D), where HSD Tukey *p*-values between 0 mM treatment and all other drug concentrations are displayed, NS: not significant. The underlying data for panels A, C, D, G, and H can be found in S1 Data. Uncropped western blots available in S1 Raw Images.

## Discussion

In this work, we performed large (phospho)proteomic studies of fission yeast sexual gamete differentiation and fusion, revealing the extensive remodeling of the (phospho)proteome during these developmental transitions. We present rich, high-confidence data sets, which document numerous phosphorylation changes likely to be driven by the activity of many kinases and phosphatases. These data sets constitute a valuable resource to understand how signaling promotes the physiology of starvation, mating, and cell fusion, which can be explored either in bulk (S1 and S2 Tables) or by browsing through the associated website (https://pombephosphoproteomics.unige.ch/).

We have set up robust synchronization procedures to conduct population-based assays of mating and fusing cells, which will be of general use for other -omics analyses. It is however important to bear in mind the specific conditions allowing optimal synchrony. First, the *cyc5Δ* cells lacking all 5 non-essential cyclins (*cig1Δ cig2Δ crs1Δ puc1Δ rem1Δ*), which extends the differentiation-permissive G1 phase and increases cell length, may mask some of the (phospho-) proteomic changes governing the starvation- and/or pheromone-dependent cell cycle arrest of wild-type cells [98,99]. Second, because of the use of the split *fus1^opto^* allele for synchronization during cell–cell fusion, we should be careful about interpretation of phosphorylation events on Fus1 and accessory proteins, as these may not represent native changes at the fusion site. Nevertheless, the few phosphorylation events that were expected (for instance that of the MAPK Spk1 active site; [55]) or independently confirmed here (Rps6) validate the approach and suggest that identified phosphorylation events also occur in wild-type strains. Third, we have used glutamate in MSL medium, instead of ammonium for pre-growth of cells to hasten differentiation, and found evidence that medium composition (MSL versus EMM) may influence signaling outcome (see discussion on autophagy below).

Our analyses of the (phospho)proteomic changes during sexual reproduction provide an interesting first global view, for which we highlight here only a few aspects. Nitrogen starvation causes massive signaling and metabolic changes, including reduction in de novo protein translation and increase in nitrogen and carbon catabolism, in part ruled by loss of TORC1 activity and increase in autophagy. Globally, these changes are very similar whether cells prepare for quiescence or differentiate sexually, as the presence of a mate modifies the starvation signaling landscape only in a relatively modest manner. The presence of a mate additionally induces signatures of G1 arrest, expression of pheromone signaling components, re-activation of growth control pathways (see below), and reorganization of the actin, polarity and cell wall machinery, likely for the morphogenetic events of shmoo outgrowth and cell fusion. A strong phospho-S/T-proline motif in phosphopeptides further suggests that many such phosphosites may be substrates of the pheromone-MAPK Spk1 though this awaits confirmation. Interestingly, some unexpected changes in differentiating haploid cells before fusion, such as expression and phosphorylation of several meiotic and sporulation factors, indicate that preparation of zygotic meiosis and sporulation already takes place in the gametes. Finally, the global analysis of (phospho)proteomic changes during cell fusion reveals an interesting inflexion at the median time of fusion, suggesting a very rapid qualitative change in signaling at the time of fusion pore opening. Thus, these data form a strong basis for future dissection of the signaling and fate changes underlying the haploid-to-diploid transition. Here, we have focused on the unexpected observation that Rps6 is highly phosphorylated during sexual differentiation and upon cell fusion.

### Reactivation of TORC1 by pheromone signaling

Our findings demonstrate that Rps6 is phosphorylated by the S6K-like Psk1 downstream of TORC1. Indeed, not only are Rps6 S235-S236 phosphorylated by Psk1 during mating, but also

Psk1 T415, which is a direct substrate of TORC1. We note that Psk1 T415 was not identified in our data sets due to absence of any peptide covering this site. Furthermore, these phosphorylation events are sensitive to rapamycin, which specifically inhibits TORC1, and this sensitivity is blocked by mutation of the TORC1 catalytic subunit Tor2. Thus, even though Rps6 is also substrate of Gad8 downstream of TORC2 [75], and TORC2 is essential for fertility [21,76,77], in wild-type cells, the critical activity change that leads to Rps6 phosphorylation during sexual reproduction is that of TORC1. In wild-type cells, our data also suggest that the activity of TORC2, although it is essential for fertility [21], may not increase during sexual reproduction, as observed from the observation that Gad8 phosphorylation does not increase. We note however that pheromone-dependent phosphorylation of Rps6 (but not Psk1) is still observed in the hypomorphic *tor2-S1837E* mutant, suggesting contribution of other kinases than Psk1 to Rps6 phosphorylation, perhaps as cellular adaptation to constitutive reduction of TORC1 activity.

At first glance, the finding that TORC1 activity increases during mating is surprising. Indeed, TORC1 is well known to monitor nutrient availability and be inactivated upon starvation. Furthermore, many lines of evidence have shown that TORC1 inactivation is both necessary and sufficient to initiate sexual differentiation in *S. pombe* [11,12,14]. For example, temperature-sensitive mutants of *tor2* lead to sexual differentiation in rich media at restrictive temperature, while mutants with hyperactive TORC1 display sterility [18,19]. Our data do not put this prior evidence in question but show that subsequent efficient progression through sexual reproduction coincides with, and requires, TORC1 reactivation.

We present several lines of evidence showing that low TORC1 activity yields low mating efficiency. Mating success was reduced in a dose-dependent manner by caffeine or auxin, molecules demonstrated to inhibit TORC1 [94–96]; mating was reduced in absence of the non-essential TORC1 subunit Tco89, whose extended lifespan and sensitivity to Torin1 and caffeine indicate low TORC1 activity [95,97]; sexual agglutination was reduced in the hypomorphic *tor2-S1837E* allele, though we did not detect reproducible reduction in overall mating success; finally, mating was also strongly reduced in *leu1-32* mutants, which we show here have reduced TORC1 activity during mating. This latter observation agrees with prior evidence of low TORC1 activity in leucine auxotrophs [92]. Notably, *leu1-* but not WT strains are sensitive to rapamycin [79] and exhibit fast onset of autophagy upon leucine depletion [91], suggesting that auxotrophs are on the verge of starvation and TORC1 inactivation. In *S. cerevisiae* and mammalian cells, leucine activates TORC1 either directly through the Leucyl-tRNA synthetase or indirectly through the Gtr1/2–RAGulator complex [89,90]. In *S. pombe*, the levels of precursor tRNA, including that for leucine, function upstream of TORC1 and are downregulated upon starvation [100], but the specific sensing pathway is unknown. We now link low TORC1 activity in *leu1-32* with poor mating and inefficient cell–cell fusion (Fig 7C and [37]). This link is further supported by the previous work on the Ppk32 kinase, deletion of which exhibits increased TORC1 activity specifically in the *leu1-32* mutant background [101]. During sexual differentiation, these cells reactivate TORC1 earlier than WT cells (a reactivation attributed to activation of autophagy in this earlier work), which restores better mating efficiency. Together, these data support the view that TORC1 activity is required for efficient mating progression.

The activity of TORC1 is likely required throughout the sexual reproduction process. Indeed, both *tor2-S1837E* and *tor2-ts10* mutants showed low spore viability. The conditional, temperature-sensitive *tor2-ts10* allele is particularly informative, as its inactivation at restrictive temperature induces sexual differentiation [14], but sexual reproduction at this restrictive temperature yields a large fraction of inviable spores, even when these are returned to permissive temperature before germination. A role for TORC1 activity during meiosis and sporulation is

also supported by the observation of meiotic defects in mutants of the Raptor-like TORC1 subunit Mip1 [102]. In fact, it makes sense that morphogenetic processes such as conjugation, cell fusion, and spore formation should require the anabolic-promoting function of TORC1. Thus, even though TORC1 activity is lost to initiate sexual differentiation, it must then be restored both in gametes and likely in the zygote to sustain sexual development.

An interesting question for the future is whether sexual development solely requires a quantitative increase in TORC1 activity or whether TORC1 signaling during mating is qualitatively different. Unfortunately, there are only few known bona fide sites in *S. pombe* besides Psk1 T415 [103]. Several TORC1 target sites have been identified on Mei2 [60], for which we recovered traces for S39, but only in 2 replicates, with a tendency to increase during mating. TORC1 was proposed to phosphorylate Atg13 [103], but specific residues are unknown. Among sites identified in at least one of the 2 previous TORC1 phosphoproteomes [49,50], sites on eIF2β (Tif212), eIF4γ (Tif471), Osh2, Rga3, Rng2, Ppk30, Bbc1, Pom1, Trs130, and a few uncharacterized proteins also increase in our mating data set, suggesting phosphorylation of Psk1 and Rps6 are not the only consequences of TORC1 reactivation. We also see increase in phosphorylation on Oca2 and Cek1, orthologues of *S. cerevisiae* Npr1 and Rim15, phosphorylated downstream of TORC1 in this organism [104]. By contrast, the main phosphorylation sites on Maf1, S63, which retards its electrophoretic migration and has been proposed as bona fide TORC1 substrate [75,105], does not show significant change in any of our data sets.

A second open question is the mechanism of TORC1 reactivation by pheromone signaling. In response to nitrogen starvation, TORC1 inactivation drives autophagy, which in turn promotes TORC1 reactivation likely by increasing intracellular amino acid levels [81–83]. We have shown that pheromone signaling does not boost autophagy, as estimated from the CTP-Atg8 cleavage assay. This result also agrees with prior data showing that ectopic activation of the sexual differentiation pathway in rich conditions does not promote autophagy [84,86]. Furthermore, even if dynamics are not identical, autophagy-deficient cells exposed to P-factor show TORC1 re-activation as do WT cells, demonstrating that pheromone-dependent reactivation of TORC1 does not require the autophagic process per se. However, because addition of low levels of nitrogen is essential for autophagy mutants to differentiate sexually, and thus to be able to respond to pheromones, we cannot know whether TORC1 re-activation by pheromones requires the availability of small amounts of nitrogen, be it delivered exogenously or by the autophagic process. We can nevertheless conclude that the dynamics of TORC1 re-activation does not follow that of nitrogen availability.

As a side note, the unexpected finding that *atg1Δ* mutants show some level (and *atg12Δ* wild-type–like levels) of CFP-Atg8 cleavage in the MSL medium suggests that autophagy is regulated by the environment. The Atg1 kinase complex sits at the top of a hierarchical autophagy pathway and integrates various inputs, including regulation by TOR kinase [106]. While it is widely considered essential for induction of autophagy, previous work has shown that the Atg1 mammalian homologues (ULK1/2) are not absolutely required for basal autophagy [107]. EMM (in which *S. pombe atg1Δ* cells are fully autophagy-deficient) and MSL are defined synthetic media with slightly different amounts of salts and ions and few additional differences. Our finding should invite examination of the role of medium composition on autophagy and more generally on cell physiology. All our experiments, except where indicated, were conducted upon starvation from glutamate-containing medium.

The activation of TORC1 by pheromone signaling is reminiscent of the activation of mTORC1 by growth factor in mammalian cells. For example, the mammalian Ras-ERK pathway promotes mTORC1 activation by phosphorylation of the Tuberin Tsc2, a GAP for the Rheb GTPase TORC1 activator [108,109]. Homologues of Rheb GTPase and its GAP exist and

control TORC1 activity in *S. pombe* cells [17,18,110,111]. It will be exciting to investigate whether the mechanism of TORC1 re-activation during mating is similar.

## Reactivation of growth promoting pathways by pheromone in nitrogen-starved cells

While we focused our attention on the reactivation of TORC1, our phosphoproteomic dataset forms a rich resource to examine the signaling changes imposed by pheromone sensing. For example, the data set provides evidence that the NDR/LATS-family Orb6 kinase is also re-activated by pheromone signaling. In animal cells, the LATS1/2 kinases are central elements of the Hippo pathway, involved in tissue growth control, and NDR1/2 kinases have been involved in multiple forms of cancer [112]. In fission yeast, the essential Orb6 kinase promotes polarized growth by antagonizing translational repression, notably of Ras GTPase activators, and by phosphorylating regulators of Cdc42 GTPase and the secretory system [48,113,114]. During N-starvation, Orb6 activity is down-regulated to extend lifespan [53]. Our data show an impressive 15-fold enrichment in Orb6 substrates among sites with phosphorylation increases during sexual differentiation relative to starvation-only conditions. Thus, like TORC1 and despite continued starvation, Orb6 is reactivated for mating, where it may promote growth of the mating projection. Thus, the 2 major growth-promoting pathways—TORC1 and Orb6—are both reactivated for sexual reproduction. It will be exciting to dissect the mechanisms by which pheromones act as growth factors.

## Materials and methods

### Yeast strains

Strains were constructed using standard genetic manipulation of *S. pombe* either by tetrad dissection or transformation and can be found in S3 Table. Plasmids generated for this study are listed in S4 Table.

To construct the optogenetic system, a homothallic (*h90*) strain, auxotroph for uracil containing a myo52-tdtomato allele at the endogenous locus was transformed by PCR-based gene targeting [115] using oligos osm933 and osm1670 on plasmid pSM693 to replace the *fus1* ORF with an hphMX cassette and generate ySM4131.

For homothallic *fus1^opto^* cells, strain ySM4131 was first transformed with pSM2470 (3′region_fus1-5′region_fus1-fus1N$^{1-793}$-CIBN-3′UTR_fus1-term$^{fus1}$-kanMX-pFA6a) linearized with BstZ17I to generate ySM4132, and then with pSM2475 (pUra4$^{AfeI}$-pfus1-cry2PHR-fus1C$^{796-1372}$-sfGFP-term$^{nmt1}$) linearized with AfeI to generate ySM4134. ySM4131 was also transformed with pSM2475 to generate a strain containing only the Cry2PHR-Fus1C-sfGFP fragment (ySM4133). To determine the timing of fusion in Fig 1C, a mTagBFP2 cytosolic marker under the *map3* P-cell–specific promoter was introduced by transforming ySM4134 with pAV0761 [116] linearized with SpeI to obtain ySM4130.

For *cycΔ5 fus1^opto^*, *h-* and *h+ cycΔ5* strains containing the *pfus1*:Cry2PHR-*fus1C*-sfGFP construct at the uracil locus were obtained through multiple crosses between yAV2050, yAV2038 [32], and ySM4130. These strains were then transformed with pSM3227 (3′region_fus1-5′region_fus1-fus1N$^{1-793}$-CIBN-3′UTR_fus1-term$^{fus1}$-hphMX-pFA6a) linearized with BstZ17I to get the *h-* and *h+ cycΔ5 fus1^opto^* strains ySM4135 and ySM4136, respectively. We used heterothallic *cycΔ5* strains to control entry into sexual differentiation, because we observed homothallic strains to mate when reaching higher densities even on nitrogen-rich media.

The *h+ cycΔ5 fus1^opto* strain (ySM4136) was further transformed with pSM3295 (pAde6^PmeI-pact1-mcherry-term^tdh1-patMX) linearized with RsrII to yield ySM4137. Colonies were selected on Edinburg Minimal Medium (EMM) supplemented with 400 μg/ml of glufosinate-ammonium (CatNo. G002P01G, Cluzeau Info Labo, Sainte-Foy-La-Grande, France) as previously described [32].

The Rps6 prototroph mutants were obtained by crossing strain ANO229 [73] to WT lab strains to generate ySM4138, ySM4139, and ySM4145.

The heterothallic *h- tor2^S1837E* strain (ySM4063) is strain TA1397 from [117]. The homothallic *h90 tor2^S1837E* strain (ySM4140) was obtained by crossing ySM4063 with a homothallic WT strain.

The homothallic *h90 psk1Δ* deletion strains (ySM4141 and ySM4142) were obtained by transforming the homothallic wild-type strain ySM1396 with pSM3571 linearized with AfeI. The homothallic *h90 psk1Δ leu1-32* strain (ySM4148) was obtained by transforming ySM4146 with pSM3571 linearized with AfeI.

The *h- sxa2Δ* prototroph strain (ySM4143) was obtained by a marker switch on the strain S16 G10 from the Bioneer library [118] to replace the kanMX6 cassette with an hphMX6 cassette, followed by crosses.

The *tor2-ts10* used in the study were obtained by crossing strain JV306 (ySM1591; *h- tor2ts-10 ade6-M216 leu1-32*) from [14] to WT homothallic strains, giving ySM4144.

Deletion of *rps602* and *rps601* in *h90 leu1-32* strains were obtained by transforming ySM4147 with pSM3297 (3′region_rps602-5′region_rps602-kanMX-pFA6a) and pSM3298 (3′region_rps601-5′region_rps601-hphMX-pFA6a) linearized with AfeI to obtain ySM4149, ySM4150 and ySM4151, ySM4152, respectively.

Deletion of *tco89* was obtained by transforming ySM1396 with pSM3364 (3′UTR_tco89-5′UTR_tco89-hphMX-pFA6a) linearized with AfeI to generate ySM4157 [116]. The heterothallic *h- CFP-atg8:leu1+ sxa2Δ::natMX6* deleted for *atg1* (ySM4157), *atg5* (ySM4174), and *atg18a* (ySM4175) strains used to test the effect of autophagy on TORC1 reactivation were obtained by transforming DY11900 (ySM4158), DY4021 (ySM4170), and DY4031 (ySM4172) (Sun and colleagues) respectively, with pSM3274 (pFA6a -3′UTR_sxa2-5′UTR_sxa2-natMX) linearized with StuI. A heterothallic *h- CFP-atg8:leu1+* strain was isolated from a cross between DY11900 and a WT strain, auxotroph for leucine and then transformed with pSM3274 linearized with StuI to obtain ySM4156.

## Time-course phosphoproteomic experiments

For time-course phosphoproteomic experiments, *h- cycΔ5 fus1^opto* and *h+ cycΔ5 fus1^opto* cells were grown and starved in a room kept in the dark and manipulation were done with the help of a red LED bulb (Osram). Unless specified, cells were always kept in the dark until they were quenched in 10% W/V TCA during the first step of protein extraction.

For the time course during sexual differentiation, precultures of *h- cycΔ5 fus1^opto* and *h+ cycΔ5 fus1^opto* were grown to late log-phase in 40 ml, diluted to 400 ml of MSL + 15 mM glutamate, and grown to an $OD_{600nm}$ = 0.5–0.9 at 30°C in the dark. For both cultures, a volume corresponding to 300 $[OD_{600nm}]$ was collected and washed separately 3 times with 20 ml of MSL-N. Centrifugation speed was set to 1,000 g for 2′ during the washes. Each strain was then resuspended in 200 ml of MSL-N and allowed to starve for 2 h at 30°C in the dark, and 100 ml of each strain were then mixed, cells were centrifuged at 1,000 g for 2′, the pellet was then resuspended in 2 ml of fresh MSL-N and 20 μl spots of the cell slurry were pipetted onto MSA-N plates at a density of 3 $[OD_{600nm}]$/spot. We noted that spotting too many cells on the same plate drastically reduced mating efficiency and for that reason, only 5 spots were

spotted on the same plate. We used an amount equivalent to 60 [$OD_{600nm}$] per time point. The remaining 100 ml of culture of each heterothallic strain was centrifuged at 1,000 g for 2′ and the pellet resuspended in 1 ml of MSL-N. Both pellets were then spotted as 20 μl spots onto separate MSA-N at a density of 3 [$OD_{600nm}$]/spot. We used an amount equivalent to 30 [$OD_{600nm}$] per strain and time point. Plates were then incubated at 30˚C in the dark until sample collection excepts the plates for the time point at 0′ which were directly used for protein extraction.

For the phosphoproteomic time course during cell–cell fusion, pre-cultures of *h- cycΔ5 fus1^{opto}* and *h+ cycΔ5 fus1^{opto}* were grown to late log-phase in 30 ml, diluted to 300 ml of MSL + glutamate and grown to an $OD_{600nm}$ = 0.5–0.9 at 30˚C in the dark. For both cultures, an amount equivalent of 150 [$OD_{600nm}$] of cells was collected and spun down at 1,000 g for 2′. The pellet was washed 3 times in 20 ml of MSL-N and re-suspended in 100 ml of MSL-N and cells were incubated for 2 h at 30˚C in the dark. Cells were then mixed together and centrifuged at 1,000 g for 2′. The pellet was then resuspended in 2 ml of fresh MSL-N and spotted as 20 μl spot onto MSA-N plates (5 spot per plate). A total of 30 [$OD_{600nm}$] were used per time point. Plates were incubated at 30˚C in dark conditions for 2 h 30′ and then shifted to light condition until sample collection. For the time point at 0 min, the plates were directly used for protein extraction without being exposed to light.

## Protein extraction and digestion

Extract protocol was adapted from [34]. Yeast cells were collected from plates using 5 ml of 10% w/v ice-cold TCA. Cells were then spun down at 1,000 g for 2′ at 4˚C and supernatant was discarded. Cells were washed first in 5 ml of acetone (cooled down at −20˚C) and then in 1 ml of lysis buffer (50 mM ammonium bicarbonate, 10 mM DTT, 5% SDS). Pellets were resuspended in 400 μl of lysis buffer. Acid-washed glass beads (Sigma; G8772) were added to the samples and cells were lysed using a FastPrep-24 5G bead beating grinder (6 times shaking at 100 V for 30", 30" break between runs). Samples were then centrifuged at max speed for 5′ at 4˚C, and supernatant was recovered as protein fraction. Samples were immediately snap-frozen. For processing, samples were thawed rapidly and heated at 95˚C for 10 min with shaking to lyse cells. Protein concentration was determined using the tryptophane fluorescence method [119]. Trypsin digestion of 120 μg of protein material per sample was carried out according to the S-TRAP (Protifi, Farmingdale, New York) method as described [120]. Briefly, after heating at 95˚C to denature and reduce disulfides, cysteines were alkylated by reaction with 30 mM (final) chloroacetamide for 1 h in the dark at RT. An aliquot of 12% phosphoric acid was added to lower pH to 3.0, followed by dilution with 4 volumes of S-TRAP loading buffer (100 mM (final concentration) triethylammonium bicarbonate buffer (TEAB) (pH 8.0), in 90% MeOH). The obtained mixture was passed by centrifugation on S-TRAP Mini cartridges, which were then washed 3 times with 600 μl of loading buffer. Digestion was started by adding to the cartridges 20 μg of Trypsin (Promega) in 125 μl of 50 mM TEAB (pH 8) and was carried out for 2 h at 47˚C without shaking. Digested peptides were eluted by centrifugation, followed by further elution of the cartridge with 80 μl of 50 mM TEAB, then 80 μl of 0.2 formic acid and 80 μl of 50% acetonitrile (each time at 3,000 × g for 1 min). All eluates were pooled, and samples were dried by evaporation.

## TMT labeling and cleanup

Dried peptide mixtures were resuspended in 45 μl of 50 mM Hepes (pH 8.3) and reacted for 1 h at room temperature with 0.4 mg of TMT 10-plex reagents (Thermo Fisher Scientific, 90110) dissolved in 27.5 μl of pure acetonitrile. The reaction was quenched by adding 7 μl of 5% (v/w)

hydroxylamine and incubating for 15 min at RT. Individual TMT labeled samples were pooled. The mixture obtained was acidified with TFA, frozen and the volume reduced to 1/10 of the initial by evaporation. After adding 9 volumes of aqueous 0.1% TFA, peptides were desalted on a C18 SepPak 1 cc 50 mg cartridge (Waters, #WAT054955). An aliquot of 10% of the eluate was dried and analyzed by MS as described below to assess labeling efficiency (which globally was found to be higher than 98.5%) and derive ratios of total protein content for normalization.

## Phosphopeptide enrichment

The remaining of the TMT mix (approx. 1.0 mg) was dried, re-dissolved, and processed for phosphopeptide enrichment by IMAC using the High-Select Fe-NTA Phosphopeptide Enrichment Kit (Thermo Fisher Scientific, A32992) according to instructions from the manufacturer. The eluate from the IMAC cartridge was dried and resuspended in 70 µl of 2% MeCN, 0.05% TFA for LC-MS/MS analysis.

## Liquid chromatography-tandem mass spectrometry

Tryptic peptide mixtures were injected on an Ultimate RSLC 3000 nanoHPLC system (Dionex, Sunnyvale, California, USA) interfaced to an Orbitrap Fusion Tribrid mass spectrometer (Thermo Scientific, Bremen, Germany). Peptides were loaded onto a trapping microcolumn Acclaim PepMap100 C18 (20 mm × 100 µm ID, 5 µm, 100 Å, Thermo Scientific) before separation on a reversed-phase custom packed nanocolumn (75 µm ID × 40 cm, 1.8 µm particles, Reprosil Pur, Dr. Maisch). A flowrate of 0.25 µl/min was used with a gradient from 4% to 76% acetonitrile in 0.1% formic acid (total method time: 140 min).

Both total proteome and phospho-enriched samples were injected multiple times with different methods and either with a normal nanospray ion source or a Field Asymmetric Waveform Ion Mobility Spectrometry interface (FAIMS pro, Thermo Fisher Scientific). For analyses with the normal nanoLC interface, a data-dependent acquisition method controlled by Xcalibur 4.2 software (Thermo Fisher Scientific) was used that optimized the number of precursors selected ("top speed") of charge 2+ to 5+ while maintaining a fixed scan cycle of 1.5 s. The precursor isolation window used was 0.7 Th. Peptides were fragmented by higher energy collision dissociation (HCD) with a normalized energy of 37% or 40% using 2 separate methods (2 serial injections). MS2 scans were done at a resolution of 50'000 in the Orbitrap cell to resolve 10-plex TMT reporter ions. The m/z of fragmented precursors was then dynamically excluded from selection during 60 s.

For analyses with the FAIMSpro interface, data-dependent acquisition methods controlled by Xcalibur 4.2 software (Thermo Fisher Scientific) were used, that alternated between 2 compensation voltages (CV) to acquire 2 survey scans within each cycle. Three methods were used, with the following CV pairs: −40/−60V, −50/−70V, −55/−65V. Following each survey scan at each CV, a "top speed" acquisition was performed to accumulate a maximum of MS2 spectra while keeping a maximum total cycle time of 1.0 s. MS2 scans were acquired with a normalized collision energy of 37%. All other parameters for MS2 spectra acquisition were the same as for the methods without FAIMS separation.

## Raw MS data analysis

Raw MS files obtained with FAIMS ion separation were split into independent files relative to each CV using the software Freestyle 1.6.90.0 (Thermo Fisher Scientific).

All tandem MS data were processed by the MaxQuant software (version 1.6.14.0) [121] incorporating the Andromeda search engine [122]. The *S. pombe* UNIPROT reference

proteome (RefProt) sequence database of March 3, 2021 was used (5,141 sequences), supplemented with sequences of common contaminants. Trypsin (cleavage at Lys, Arg) was used as the enzyme definition, allowing 2 missed cleavages. Carbamidomethylation of cysteine was specified as a fixed modification. N-terminal acetylation of protein, oxidation of methionine and phosphorylation on Ser, Thr, and Tyr were specified as variable modifications. All identifications were filtered at 1% FDR at both the peptide and protein levels with default MaxQuant parameters. The isobaric match between runs functions of MaxQuant was used [123]. For comparison of TMT runs, ratios were automatically calculated by MaxQuant as a function of the reference channels [123].

MaxQuant data were further processed with Perseus software (version 1.6.15.0) [124] for the filtering, log2-transformation and normalization of values. Medians of TMT ratios across samples obtained from the total protein measurements were used to correct ratios for phosphoproteomics data to account for differences in total loaded material.

## Statistical analysis

Prior to statistical analysis, to correct for technical variation between replicates, for all phosphosites and protein identified, the median of all time point values collected for a given replicate was subtracted from individual values. In the time-course phosphoproteomic study during sexual differentiation, all 10 values (the 5 values from time points during starvation and 5 values from time points during sexual differentiation) were used to calculate the median. In the time-course phosphoproteomic study during cell–cell fusion, all 5 time point values were used. This method gave good stratification of starvation and mating data in principal component analysis (Figs 3B and 4B).

Sites that were absent in more than 1 biological replicate were discarded: in the time course during sexual differentiation, we only kept sites with no more than 10 missing values out of 30 total possible values (10,828 sites from 1,884 distinct proteins were retained); in the time course during cell–cell fusion, we only kept sites with no more than 5 missing values out of 15 total possible values (11,979 sites from 2,065 distinct proteins). In the raw data, a few sites have exactly the same numeric values across all measurements, because they originate from sites with multiple phosphorylation possibilities. To not bias the statistical multiple correction, redundancy was removed and only one of each value was kept for the statistical analysis ($n$ = 9,951 for the sexual differentiation, $n$ = 11,286 for the cell–cell fusion). Duplicate sites were added to the results a posteriori.

The normalized data was used to fit a linear model with the R Package "limma" [125] to identify sites behaving differently during the time course using the eBayes function with default parameters.

The following contrasts were used:

For starvation:

Ctrl_45–0 = (Ctrl.45min-Ctrl.0min)

Ctrl_90–45 = (Ctrl.90min-Ctrl.45min)

Ctrl_135–90 = (Ctrl.135min-Ctrl.90min)

Ctrl_180–135 = (Ctrl.180min-Ctrl.135min)

For mating:

Diff_45–0 = (Mating.45min-Mating.0min)-(Ctrl.45min-Ctrl.0min)

Diff_90–45 = (Mating.90min-Mating.45min)-(Ctrl.90min-Ctrl.45min)

Diff_135–90 = (Mating.135min-Mating.90min)-(Ctrl.135min-Ctrl.90min)

Diff_180–135 = (Mating.180min-Mating.135min)-(Ctrl.180min-Ctrl.135min)

Subtraction of starvation signal to mating signal was used to eliminate sites that vary with a similar slope in the starvation and mating samples. Phosphosites that vary in both conditions but in a different manner (either same direction but different slopes, or increase versus decrease) will be retained (S2A Fig). We note that the strategy may slightly underestimate the true extent of changes during mating, because possible addition of noise may increase variability between replicates and reduce statistical significance in a few cases.

A total of 262 significant sites (Benjamini–Hochberg adjusted $p$-value $\leq 0.05$) were identified as significantly different between the control and mating time course. Sites with the same original measurements as the significant ones that were excluded because they were redundant (multiple possibilities of phosphorylation in the fragment detected by the mass spectrometer) were added back, leading to a total of 272 candidate sites.

The parameters for the linear model used in the cell–cell fusion time course were:

Diff_11–0 = (Fusion.11min-Fusion.0min)

Diff_22–11 = (Fusion.22min-Fusion.11min)

Diff_33–22 = (Fusion.33min-Fusion.22min)

Diff_44–33 = (Fusion.44min-Fusion.33min)

A total of 428 significant sites (Benjamini–Hochberg adjusted $p$-value $\leq 0.001$) were identified as significantly varying during the fusion time course. We used a lower adjusted $p$-value than for the sexual differentiation time course because this experiment had 4 biological replicates (3 for the differentiation time course). Sites with the same original measurements as the significant ones that were excluded because they were redundant (multiple possibilities of phosphorylation in the fragment detected by the mass spectrometer) were added back, leading to a total of 440 candidate sites.

For proteomics analysis, MS2 spectra from unique peptides and razor peptides (assigned peptides to the protein with the highest Protein IDs among all possible proteins) were considered for protein quantification. Intensities at each time point are the sums of all individual peptide intensities belonging to a particular protein group. Unique and razor peptide intensities are used as default.

For analysis of covariation, we first selected all the phosphosites varying significantly during starvation, sexual differentiation, and cell–cell fusion for which the protein had also been identified to change significantly in the proteomic data set. Then, log2 fold-changes values were used to fit a multiple linear regression model with interaction between a categorical variable (Proteomics versus Phosphoproteomics) and a continuous variable (time). The $p$-values for the interaction term were corrected for multiple testing using a Benjamini–Hochberg correction and phosphosites with $p$-values higher than 0.05 were considered as co-varying. If there was no significant change on protein levels, phosphosite changes were considered not to be co-varying.

Gene ontology enrichment was done using PANTHER (Version 16.0/17.0). For the phosphoproteomic time course during mating, phosphosites significantly changing during starvation or mating (after accounting for changes during starvation) were first divided into 2 groups: increasing or decreasing phosphorylation. The list containing the Uniprot accession number of all the proteins for each group was analyzed with the PANTHER classification

system [126]. *P*-values of gene ontology were corrected for false discovery rates. Due to redundancy between some gene ontology terms, only a few significant terms are displayed.

For kinase substrate enrichment, substrates identified in previous phosphoproteomic studies were used, keeping thresholds and criteria applied in these earlier studies. Specifically, Table S1 from [34] was used to define Cdc2 substrate, table EV3 from [50] was used for TORC1 and TORC2 substrates, S15 and S16 Tables from [49] were used for TORC1 and TORC2 substrates, respectively, and S2 Table from [48] was used for Orb6 substrates. Fisher exact test was used to test for enrichment in kinase substrate during mating and starvation. For simplicity, only phosphosites detected on monophosphorylated phosphopeptides were considered for this analysis.

For motif analysis, the sequences of all the phosphosites significantly increasing during mating were aligned from the amino acid at position -5 from the phosphoresidue to the amino acid at position +5. The logo was obtained using the R package "ggseqlogo" [127] with the method parameter set to "prob" (probability). To identify new potential motifs, proline-directed sites and Orb6 substrates were removed, and the motif was plotted with the remaining phosphosites.

Two-tailed two-sample *t* tests were used for statistical analysis on Figs 1E, 7A, 7C, 7H, and S1C. The effect of the length of the incubation in the dark on the fusion efficiency of *fus1*$^{opto}$ cells (Fig 1E), and of Caffeine and Auxin treatment on wild-type cells during mating (Fig 7D) were analyzed by one-way ANOVA. Post hoc Tukey tests were performed and the *p*-values for comparisons between the condition without the drug and all other conditions are displayed.

## Strains, mediums, and growth conditions for microscopy

Protocols described in [39] were used with minor adaptation. Strains were always pre-cultured in 3 ml of MSL supplemented with appropriate amino acid and nitrogen source overnight at 30°C. Cultures were then diluted in the morning to $OD_{600nm} = 0.2$ and in the evening in 20 ml of medium at $OD_{600nm} = 0.025–0.04$. Strains used for experiments shown in Figs 1B, 1C, 1D, 1E, 7C, 7D, and 7H were grown in MSL supplemented with 15.1 mM of ammonium sulfate (MSL+N) and all other experiments were done using MSL + 15.1 mM glutamate, unless stated otherwise on the figure panel. The next morning, 4.5 $[OD_{600nm}]$ of cells were washed 3 times in 1 ml of MSL-N (centrifugation steps: 1,000 g for 2') and either spotted onto MSA-N plates (Figs 1D, 1E, 2D, S1A and S1B), incubated in liquid MSL-N (Figs 2A, 2C, 7C, 7D, 7H, and S1) or incubated in liquid MSL-N before transfer to a MSL-N 2% agarose pad (Figs 1B, 1C, 2B, and 7A). Cells were generally grown at 30°C, except for data presented in Fig 7A, 7C, 7D, and 7H, where cells were grown at 25°C. For quantifications in Fig 7D, caffeine and auxin (IAA; Indole-3-acetic acid), or DMSO were added when cultures were shifted to MSL-N. Caffeine (Cayman chemical, 14118) was freshly dissolved at 10 mM concentration in MSL-N liquid medium and diluted to reach indicated concentrations; IAA (Merck, 1.00353.0010) was dissolved in DMSO and added to cultures from a 1,000× concentrated stock. For quantifications in Figs 2C, 2D, S1A and S1B, heterothallic *h*+ and *h*- cells were starved separately for 2 h in MSL-N liquid at 30°C prior to cell mixing. The time point zero corresponds to the time the 2 mating types are resuspended together in MSL-N liquid (Fig 2C) or on MSA-N plate (Fig 2D). Except where specified, cells were imaged after 24 h.

For imaging from a pad (Figs 1B, 1C, 2B, and 7A), 100 µl of cells were taken after 3 to 4 h of incubation in liquid MSL-N if cells were grown in MSL + ammonium sulfate (Fig 1B and 1C), 0 to 1 h of incubation in liquid MSL-N if cells were grown in MSL + glutamate (Figs 2B and 6A) and centrifuged at 1,000 g for 2′. Approximately 90 µl of supernatant was removed and cells were resuspended in the remaining medium. A drop of 1 µl of the cell slurry was spotted

onto MSL-N + 2% agarose pad between a slide and a coverslip, sealed with VALAP (Vaseline: Lanolin:Paraffin, 1:1:1), essentially as described [39].

The MSL + glutamate, MSL + proline, and MSL + phenylalanine media used for experiment in Fig 2A were prepared similarly to the MSL+N [39] medium with the replacement of the 15.1 mM of ammonium sulfate with 15.1 mM of L-Glutamic acid monosodium monohydrate (Fluka, 49621), L-Phenylalanine (Sigma-Aldrich, P2126), or L-Proline (Sigma-Aldrich, P8865).

## Spore viability assay

To assess spore viability (Fig 7F and 7G), *h90* strains were spotted on MSA-N plates and incubated for 2 days at 25˚C or 32˚C. For Fig 7F, cells were resuspended in $H_2O$ and spread on YE plates. Tetrads were selected by using a tetrad micro dissector microscope (MSM400, Singer Instruments) to place 1 whole tetrad at each position. Plates were incubated for 5 days at 25˚C, before taking images. For Fig 7G, cells were resuspended in $H_2O$ containing glusulase (from 100× stock, PerkinElmer, NEE154001EA) and incubated over night at 30˚C. Spores were counted using a Neubauer counting chamber and 200, 100, and 50 spores were spread on duplicate YE plates. Plates were incubated for 4 days at 25˚C, before counting colonies. We noted a systematic error in spore counting where an average of 192 colonies germinated for 100 counted WT spores at 25˚C. We assumed this was due to imprecision in the volume of the counting chamber and corrected all values by 1.92. The graph therefore represents the "% corrected spore viability".

## Microscopy and quantification

Images shown and used for quantifications in Figs 1B, 1C, 1D, 1E, 2A, 2B, 7A, 7C, 7D and 7H, were obtained using a DeltaVision platform (Applied Precision) composed of a customized inverted microscope (IX-71; Olympus), a UPlan Apochromat 100×/1.4 NA oil objective, a camera (4.2Mpx PrimeBSI sCMOS camera; Photometrics), and a color combined unit illuminator (Insight SSI 7; Social Science Insights). Images were acquired using softWoRx v4.1.2 software (Applied Precision). For optogenetic experiment, a 550 nm longpass filter (Thorlabs, Newton, New Jersey, USA) was installed on the condenser to allow for cell visualization without photoactivation before imaging. For Fig 1B, cells were imaged every 4′, with 150 ms exposure time in the mcherry channel, 15 ms exposure in the GFP channel, and 80 ms exposure in the DAPI channel. The images were first taken in the mcherry channel (575 nm laser excitation) to record an event before photo-activation.

Images taken to optimize conditions for synchronous mating were acquired with a Leica DMI4000B microscope equipped with a standard mercury lamp and a sCMOS Pco edge 5.5 camera (Figs 2C, 2D, and S1C). Images were acquired using micromanager. Bright field images were used to quantify mating efficiency. Fusion efficiency was quantified in the red channel (excitation filter (546/12), dichroic mirror (560), emission filter (605/75)) from a mix of non-fluorescent *h- cycΔ5* cells and *h+ cycΔ5* cells expressing mcherry under the $p^{act1}$ promoter. Pairs were considered fused if the mcherry fluorescence were present in both cells of a mating pair.

The mating efficiency (% mating) represents the % of cells engaged in mating (cell pairs or fused zygotes) relative to the total number of cells:

$$mating\ efficiency = \frac{2(mating\ pairs + zygotes)}{individual\ cells\ +\ 2(mating\ pairs + zygotes)}\ X\ 100$$

The fusion efficiency (% fusion) represents the % of fused pairs (zygotes) relative to cells engaged in mating (cell pairs and zygotes):

$$fusion\ efficiency = \frac{fused\ pairs}{mating\ pairs}\ x\ 100$$

In Fig 2D, we also show the % of zygotes relative to the total number of cells:

$$\%\ fusion = \frac{2(zygotes)}{individual\ cells\ +\ 2(mating\ pairs\ +\ zygotes)}\ X\ 100$$

## Experimental conditions and protein extraction for western blot

Pre-cultures in 10 ml of MSL + 15 mM glutamate were incubated overnight at 30˚C on a rotative platform (180 rpm), diluted the next morning to $[O.D.]_{600nm} = 0.2$, in the evening to $[O.D.]_{600nm} = 0.02–0.04$ in 100 ml of MSL + 15 mM glutamate for homothallic strains and 50 ml of MSL + 15 mM glutamate for heterothallic strains. The morning of the third day, 60 $[O.D.]_{600nm}$ of cells were washed 3 times in 10 ml of MSL-N, resuspended in 40 ml MSL-N, and incubated at 30˚C for 6 to 7 h before protein extraction. *cycΔ5* cells were only incubated 3 h in MSL-N as sexual differentiation is rapid in this mutant. For experiment with Torin1 and Rapamycin (Figs 5B, 5C, and S6A), strains were incubated in MSL-N for 6 h 15 and 40 ml of cells were then treated with either 40 µl of Torin1 (5 mM stock in DMSO; final concentration 5 µm) (LC Laboratories; T-7887), 40 µl of Rapamycin (300 µg/ml stock in DMSO; final concentration 300 ng/ml) (LC Laboratories; R-5000), or 40 µl of DMSO and incubated for an additional 30 min at 30˚C. We found that higher doses of Torin1 were required for loss of Rps6 and Psk1 phosphorylation during mating (S5A Fig).

The heterothallic *sxa2Δ* time course shown in Fig 5E in the presence of pheromones was performed as followed: heterothallic *h- sxa2Δ* cells were grown in 200 ml of MSL + 15.1 mM glutamate overnight at 30˚C, diluted the next morning to $[O.D.]_{600nm} = 0.2$, diluted to $[O.D.]_{600nm} = 0.02–0.04$ in the evening in 1 L of MSL + 15.1 mM glutamate and incubated at 30˚C. In the morning of the third day, 60 $OD_{600}$ of the culture were first spun down at 1,000 g for 2′ and resuspended in 40 ml of MSL+15.1 mM glutamate prior to protein extraction. This represents time point 0. An additional 480 $[O.D.]_{600nm}$ (60 $[O.D.]_{600nm}$ per time point and condition) of cells were collected, washed 3 times in 10 ml MSL-N, and resuspended in 320 ml MSL-N. The culture was then split in 2: one culture was treated with 160 µl of P-factor (Schafer-N) (from a 1 mg/ml stock in MetOH), the second culture was treated with the same volume of MetOH. The cultures were incubated at 30˚C on a rotative platform shaking at 180 rpm; 40 ml of each culture were taken at each indicated time point for protein extraction.

The time-course experiments on heterothallic *h- sxa2Δ* WT and autophagy-deficient mutants (Figs 6B, 6F, S7C and S7D) were performed similarly to the experiment shown in Fig 5E, with 2 exceptions: (1) MSL-N was replaced by MSL + 0.5 mM glutamate (for WT and the *atg1Δ* mutant) or MSL + 0.75 mM glutamate (for WT and the *atg5Δ* and *atg18aΔ* mutants); (2) instead of resuspending the cells in MSL-N medium, 60 $[O.D]_{600nm}$ of cells were plated onto MSA + 0.5 mM glutamate (for the *atg1Δ* experiment) or MSA + 0.75 mM glutamate (for the *atg5Δ* and *atg18aΔ* experiments), supplemented with 1 µg/ml of P-factor or the equivalent volume of MetOH. The choice of MSA over MSL is due to the observation that autophagy-deficient mutants did not respond to pheromone in the liquid media. We also observed that plating 60 $[O.D.]_{600nm}$ of cells on thick agar plates (made with 40 ml of medium) promoted a better response to P-factor in the autophagy-deficient strains compared to using plates made

with 20 ml of medium. This may be due to the increased overall amount of glutamate available for shmoo formation.

For the CFP-Atg8 cleavage assays (Figs 6B, 6C, 6D, S7B S7C and S7D), cells were grown in indicated medium (MSL or EMM with indicated nitrogen source) overnight at 30°C, diluted the next morning to $[O.D.]_{600nm}$ = 0.2, diluted to $[O.D.]_{600nm}$ = 0.02–0.04 in the evening, and incubated at 30°C. In the morning of the third day, 60 $[O.D.]_{600nm}$ per sample of each strain were washed 3 times in 10 ml nitrogen-free medium, resuspended in 40 ml of the same medium−N, and incubated at 30°C on a rotative platform set to 180 rpm for indicated time prior to protein extraction.

For protein extraction, cultures were quenched by adding 100% w/v ice-cold TCA to a final concentration of 10%, centrifuged at 4°C at 1,000 g for 2′ and the supernatant was removed. The pellet was washed first with 5 ml of −20°C acetone, then with 1 ml of western blot buffer (2% SDS, 5% Glycerol, 50 mM Tris-HCl, 0.2 M EDTA + cOmplete protease inhibitor cocktail tablet (Roche, 11697498001) + PhosSTOP (Roche, 4906837001)). Pellet was resuspended in 400 μl of western blot buffer. Acid-washed glass beads (Sigma, G8772) were added to the samples and cells were lysed using a FastPrep-24 5G bead beating grinder (6 times shaking at 100 V for 30", 30" break between runs). Samples were then centrifuged at max speed for 5′ at 4°C, and supernatant was recovered as protein fraction. Samples were immediately snap-frozen. Protein samples were incubated in NuPAGE LDS sample buffer and denatured 10′ at 65°C; 1 μl of β-mercaptoethanol was added per 20 μl of sample and samples were incubated 10′. Samples were then either stored at −20°C or used directly for SDS-PAGE.

## Western blotting

A total of about 75 ng of proteins were loaded on 4% to 20% acrylamide gels (GenScript, M00655) and run at 120 V for 90′ in commercial Tris-MOPS-SDS running buffer (GenScript, M00138). Transfer was done on a nitrocellulose membrane (Sigma, GE1060001) in homemade transfer buffer (50 mM tris base, 38 mM Glycine, 1% SDS) supplemented with 20% EtOH at 100 V for 90′. Membranes were blocked 1 h in TBST + 5% milk and incubated overnight in TBST + 5% milk + primary antibodies. Phospho-Akt substrate Rabbit monoclonal antibodies (1:1,000, Cell Signaling Technology, 9614) were used to detect phospho-Rps6. Rps6 levels were detected using anti-rps6 rabbit monoclonal antibodies (1:1,000, Abcam, ab40820). Psk1 phosphorylation was detected using phospho-P70 S6K mouse monoclonal antibodies (1:1,000, Cell Signaling Technology; 9206). Psk1 levels were detected with anti-Psk1 (1:5,000) antibodies [128]. Membranes were washed 3 times in TBST for 15′ per wash and incubated 1 h in HRP-conjugated anti-rabbit (Promega, W4011) or anti-mouse (Promega, W402B) secondary antibodies (1:3,000) in TBST+ 5% Milk. Membranes were washed 3 additional times in TBST for 15′ per wash. Membranes were then covered with a mix of 1 ml of Reagent A and 1 ml of Reagent B from a Pierce ECL western blotting substrate kit (Thermo Fisher, 32106). Excess liquid was removed, and membranes were placed in transparent plastic sheets. Chemiluminescence was revealed using an Amersham imager 680 (GE Healthcare Life Sciences). Uncropped western blots are shown in S1 Raw Images.

## Supporting information

**S1 Fig.** *cycΔ5* **strains mate efficiently and Fus1N fusogenic activity is negligible at short time frames.** (**A, B**) Mating efficiency (% of cell pairs among all cells; A) and fusion efficiency (% of paired cells that fused; B) of WT or *cycΔ5* cells after 24 h on plates lacking nitrogen. The cells were pre-grown on ammonium or glutamate-containing medium. (**C**) Fusion efficiency (% of paired cells that fused) of *h90 fus1^{opto}* cells starved and kept in the dark for 24 h (same

data as on Fig 1D) and of *h+* x *h- cycΔ5 fus1^{opto}* cells individually starved for 2 h and then mixed and kept in the dark for a further 2 h 30. The latter *h+* x *h- cycΔ5 fus1^{opto}* conditions are identical to those used in the (phospho)proteomic time course, corresponding to time 150 min in Fig 3A. As these cells do not fuse, this confirms that all phosphorylation changes observed in the mating time course occur before gamete fusion. *T* test *p*-value is indicated. The underlying data can be found in S1 Data.
(PDF)

**S2 Fig. Reproducibility of the data sets and expected phosphorylation sites.** (**A**) Example profiles of phosphosites in the 3 biological replicates of the starvation/mating time course and the 4 biological replicates of the cell–cell fusion time course. Starvation (cells plated on distinct plates), mating (h+ and h- partners plated together), and starvation-corrected mating values (mating—starvation) are shown in the first 3 columns. The cell–cell fusion time course is shown in the last column. The first 3 examples (Spk1, Rps601, and Rec10) show examples of sites that do not vary significantly upon starvation but change during mating and cell–cell fusion. The next example (Ght1) shows a site that varies during starvation similarly whether cells are starved separately or in presence of their mating partner, resulting in absence of change in the starvation-corrected (mating–starvation) values, indicating no specific change during mating. The last 2 examples show sites that vary significantly both during starvation and mating, but with distinct dynamics. In the Mei2 example, variation occurs in opposite direction. In the Fur4 example, the slope of the change is distinct. Red line indicates average; black lines indicate individual replicates. Pale graphs indicate absence of significant change. The underlying data can be found in S1 and S2 Tables. (**B**) Short list of expected and identified phosphorylation increases during mating. References are to the expected sites.
(PDF)

**S3 Fig. Analysis of proteomic changes during N-starvation.** Changes in the proteome of heterothallic cells in a time course of nitrogen starvation starting at t0 = plating of cells on MSL-N plates 2 h after transfer to liquid MSL-N. (**A**) Heatmap of the significant changes in the levels of 847 proteins during nitrogen starvation, showing 2 major clusters of proteins whose level increase (1) or decrease (2). The underlying data can be found in S1 Table. (**B**) Significant fold enrichment in GO annotations for biological processes, molecular functions, and cellular components of proteins whose level increases during nitrogen starvation. (**C**) Significant fold enrichment in GO annotations for biological processes, molecular functions, and cellular components of proteins whose level decreases during nitrogen starvation. Significance levels were assessed by Fisher's exact test and corrected for false discovery rate.
(PDF)

**S4 Fig. Analysis of phosphoproteomic changes during N-starvation.** Changes in the phosphoproteome of heterothallic cells in a time course of nitrogen starvation starting at t0 = plating of cells on MSL-N plates 2 h after transfer to liquid MSL-N. (**A**) Heatmap of the significant changes in 3,100 phosphosites during nitrogen starvation, showing 2 major clusters of sites whose phosphorylation increase (1) or decrease (2). Corresponding protein dynamics is shown on the right with missing values in gray. Regression analysis estimated that about 78% of phosphosite level changes are independent of the changes in protein levels (see Fig 3C). The underlying data can be found in S1 Table. (**B**) Significant fold enrichment in GO annotations for biological processes, molecular functions, and cellular components of proteins containing one or several sites showing phosphorylation increase during nitrogen starvation. (**C**) Significant fold enrichment in GO annotations for biological processes, molecular functions, and cellular components of proteins containing one or several sites showing phosphorylation

decrease during nitrogen starvation. Significance levels were assessed by Fisher's exact test and corrected for false discovery rate.
(PDF)

**S5 Fig. Analysis of proteomic and phosphoproteomic changes during sexual differentiation.** Changes in the proteome and phosphoproteome of mating cells in a time course starting at t0 = cell mixing on MSL-N plates in the dark. The cells were pre-grown separately in liquid MSL-N for 2 h. All Data are corrected for changes happening during starvation in absence of a mating partner. (**A**) Heatmap of the significant changes in the levels of 54 proteins during mating, showing 2 major clusters of proteins whose level increase (1) or decrease (2). The underlying data can be found in S1 Table. (**B**) Significant fold enrichment in GO annotations for biological processes, molecular functions, and cellular components in proteins whose level increases during mating. No significant enrichment was found for the few proteins whose levels decrease. (**C**) Significant fold enrichment in GO annotations for biological processes, molecular functions, and cellular components of proteins containing one or several sites showing phosphorylation increase during nitrogen starvation. (**D**) Significant fold enrichment in GO annotations for biological processes, molecular functions, and cellular components of proteins containing one or several sites showing phosphorylation decrease during nitrogen starvation. Significance levels were assessed by Fisher's exact test and corrected for false discovery rate. (**E**) Analysis of the protein sequence surrounding serine and threonine residues phosphorylated during mating. The height of the amino acid one-letter code represents the frequency. The logo on the left was made using all 192 increasing phosphosites. For the one on the left, all proline-directed sites and Orb6 substrates were removed, leaving 122 phosphosites.
(PDF)

**S6 Fig. Analysis of proteomic and phosphoproteomic changes during cell–cell fusion.** Changes in the proteome and phosphoproteome in a time course starting at t0 = light exposure. *h-* and *h+ cycΔ5 fus1^opto* cells were pre-grown separately in liquid MSL-N for 2 h, mixed on plates for 150 in the dark before illumination. (**A**) Heatmap of the significant changes in the levels of 41 proteins during mating, showing 2 major clusters of proteins whose level increase (1) or decrease (2). Note that cluster 1 can be separated in 2 subclusters with levels increase at 11 or 33 min. The underlying data can be found in S1 Table. (**B**) Significant fold enrichment in GO annotations for cellular components of proteins whose level decreases during mating. No significant enrichment was found for the few proteins whose levels increase. (**C**) Significant fold enrichment in GO annotations for biological processes and cellular components of proteins containing one or several sites showing phosphorylation increase during cell–cell fusion. (**D**) Significant fold enrichment in GO annotations for biological processes, molecular functions, and cellular components of proteins containing one or several sites showing phosphorylation decrease during nitrogen starvation. Significance levels were assessed by Fisher's exact test and corrected for false discovery rate.
(PDF)

**S7 Fig. Additional western blots and controls.** (**A**) Phospho-Rps6 and phospho-Psk1 levels *in WT* heterothallic *(*h-*)* and homothallic *(*h90*)* cells starved for 6 h 15 and then treated with Torin1 (5 or 25 μm) or DMSO (0) for 30 min. (**B**) CFP-Atg8 cleavage in WT and *atg1Δ* mutants after 4 h and 8 h of nitrogen starvation in MSL. The cells were pre-grown in MSL + glutamate (top) or MSL + ammonium (bottom). (**C**) Time course of phospho-Psk1 in *h-sxa2Δ atg18aΔ* cells transferred to MSL + 0.75 mg/ml glutamate in the presence of P-factor (1 μg/ml) or MetOH at T = 0 min. CFP-Atg8 cleavage is shown in the bottom blot. (**D**) Time course of CFP-Atg8 cleavage in *h- sxa2Δ* cells, which are otherwise WT (top), *atg1Δ* (middle),

or *atg5Δ* (bottom), transferred to MSL + 0.75 mg/ml (or 0.5 mg/ml in case of *atg1Δ*) glutamate in the presence of P-factor (1 μg/ml) or MetOH at T = 0 min. These are the same extracts as probed for phospho-Psk1 in Fig 6F. Uncropped western blots available in S1 Raw Images.
(PDF)

**S1 Table. (Phospho)proteomics data for all proteins and sites with significant change.** All proteins and phosphosites whose levels are significantly changing during starvation, mating, or fusion time course. The data are presented as 6 distinct tabs in.xls format, presenting proteomics and phosphoproteomics over the starvation time course, proteomics and phosphoproteomics over the mating time course, and proteomics and phosphoproteomics over the fusion time course. Log2-transformed, median-corrected values are shown for all, except for the mating time course where the starvation values have been subtracted. Lighter shading indicates data from individual replicates; darker shading shows average values. An additional first tab presents a detailed legend.
(XLSX)

**S2 Table. Complete (phospho)proteomics data.** All proteins and phosphosites identified in during starvation, mating, or fusion time course. Data presented in S1 Table is also included. The data are presented as 4 distinct tabs in.xls format, presenting proteomics and phosphoproteomics over the starvation and mating time course, and proteomics and phosphoproteomics over the fusion time course. Log2-transformed, median-corrected values are shown for all. Lighter shading indicates data from individual replicates; darker shading shows average values. In the starvation-mating time course, different color shadings highlight the starvation data, the uncorrected mating data and the starvation-subtracted mating data. An additional first tab presents a detailed legend.
(XLSX)

**S3 Table. Strains used in this study.**
(PDF)

**S4 Table. Plasmids used in this study.**
(PDF)

**S1 Raw Images. Uncropped western blots.** For each figure panel, the panel is shown for reference. Each of the full-size, uncropped western blots is shown by itself and overlaid with the ladder. The relevant protein is indicated with an arrowhead. Asterisks mark lanes not included in the final figures.
(PDF)

**S1 Data. Data set.**
(XLSX)

## Acknowledgments

We thank Dr. Ronit Weisman (Open University of Israel), Dr. Masayuki Yamamoto (NIBB), Dr. Fuyuhiko Tamonoi (UCLA), and Dr. Li-Lin Du (NIBS) for strains; Dr. Ronit Weisman and Dr. Kazuhiro Shiozaki (NIST) for antibodies; Dr. Manfredo Quadroni and members of the Protein Analysis Facility (UNIL) for the mass-spectrometry analysis; Dr. Nicolas Guex at the Bioinformatics Competence Center (UNIL) for the linear model statistical analysis of the (phospho)proteomic data; Sajjita Saha (UNIGE) for help with setting up the conditions for the autophagy mutants; Nicolas Roggli at the Department of Molecular and Cellular Biology (UNIGE) for preparation of the (phospho)proteome website; and Dr. Robbie Loewith

(UNIGE) for discussions and comments on the manuscript. The project was designed by SGM and MB. LM performed experiments shown in Fig 7C–7H and helped with Western blotting in Figs 5 and 6. MB performed all other experiments. MB made all the analysis of the (phospho)proteomic data. SGM wrote the paper, with revisions from all authors. SGM provided supervision and acquired funding.

## Author Contributions

**Conceptualization:** Melvin Bérard, Sophie G. Martin.

**Data curation:** Melvin Bérard.

**Formal analysis:** Melvin Bérard, Laura Merlini, Sophie G. Martin.

**Investigation:** Melvin Bérard, Laura Merlini.

**Methodology:** Melvin Bérard, Laura Merlini.

**Project administration:** Sophie G. Martin.

**Supervision:** Sophie G. Martin.

**Validation:** Melvin Bérard.

**Visualization:** Melvin Bérard, Sophie G. Martin.

**Writing – original draft:** Sophie G. Martin.

**Writing – review & editing:** Melvin Bérard, Laura Merlini, Sophie G. Martin.

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
