## [Editor Report · Decision Letter 0]

2 Oct 2024

Dear Sophie, 

Thank you very much for submitting your revised manuscript from Review Commons entitled "TORC1 reactivation by pheromone signaling revealed by phosphoproteomics of fission yeast sexual reproduction" for consideration as a Methods and Resources Article by PLOS Biology.

Your manuscript has now been evaluated by the PLOS Biology editorial staff, as well as by an academic editor with relevant expertise, and I am writing to let you know that we would like to send your revised submission back to the original reviewers at Review Commons.

Once your full submission is complete, your paper will undergo a series of checks in preparation for peer review. After your manuscript has passed the checks it will be sent out for review. To provide the metadata for your submission, please Login to Editorial Manager (https://www.editorialmanager.com/pbiology) within two working days, i.e. by Oct 04 2024 11:59PM.

Best wishes,

Richard

Richard Hodge, PhD

rhodge@plos.org

PLOS

---

## [Decision Letter · Decision Letter 1]

12 Nov 2024

Dear Sophie,

Thank you for your patience while your revised manuscript from Review Commons entitled "TORC1 reactivation by pheromone signaling revealed by phosphoproteomics of fission yeast sexual reproduction" was peer-reviewed at PLOS Biology. Please accept my sincere apologies for the delays that you have experienced during the re-review process. Your manuscript has now been evaluated by the PLOS Biology editors, an Academic Editor with relevant expertise, and by the original reviewers at Review Commons.

Based on the reviews, I am pleased to say that the reviewers are satisfied with the responses to their previous comments and we are likely to accept this manuscript for publication. Reviewer #2 provides several suggestions to improve the readability of the text that we encourage you to include. In addition, I would be grateful if you could please make sure to address the following data and other policy-related requests that I have provided below (A-H):

(A) We would like to suggest the following minor modification to the title:

“Proteomic and phosphoproteomic analyses reveal that TORC1 is reactivated by pheromone signaling during sexual reproduction in fission yeast”

(B) After some discussions with the wider editorial team and the Academic Editor, we think that your manuscript would be a better fit as a Research Article at the journal. Whilst we appreciate the Resource value of the proteomic and phosphoproteomic analyses and understand why the ‘Methods and Resources’ article was initially chosen, we feel that the subsequent TORC1 and pheromone signaling story lends itself more to a conventional Research Article. We will change the article type on your behalf when you submit the revision if you are OK with this way forward.

(C) You may be aware of the PLOS Data Policy, which requires that all data be made available without restriction: http://journals.plos.org/plosbiology/s/data-availability. For more information, please also see this editorial: http://dx.doi.org/10.1371/journal.pbio.1001797

-Supplementary files (e.g., excel). Please ensure that all data files are uploaded as 'Supporting Information' and are invariably referred to (in the manuscript, figure legends, and the Description field when uploading your files) using the following format verbatim: S1 Data, S2 Data, etc. Multiple panels of a single or even several figures can be included as multiple sheets in one excel file that is saved using exactly the following convention: S1_Data.xlsx (using an underscore).

-Deposition in a publicly available repository. Please also provide the accession code or a reviewer link so that we may view your data before publication. 

Figure 1C-E, 2A, 2C, 2E, 3B, 3D-I, 4B, 4D-H, 7A, 7C-D, 7G-H, S1A-C, S2, S3A-C, S4A-C, S5A-E, S6A-D

(D) Please deposit the proteomic and phosphoproteomic datasets in a public data repository such as PRIDE (https://www.ebi.ac.uk/pride/). Please ensure that the data is made publicly available and provide the accession number in the Data Availability Statement in the online submission form. 

(E) Please also ensure that each of the relevant figure legends in your manuscript include information on *WHERE THE UNDERLYING DATA CAN BE FOUND*, and ensure your supplemental data file/s has a legend.

(F) We require the original, uncropped and minimally adjusted images supporting all blot and gel results reported in the following Figures:

Figure 5A-F, 6B-D, 6F, 7B, S7A-D

We will require these files before a manuscript can be accepted so please prepare and upload them now. Please carefully read our guidelines for how to prepare and upload this data: https://journals.plos.org/plosbiology/s/figures#loc-blot-and-gel-reporting-requirements

(G) Per journal policy, if you have generated any custom code during the course of this investigation, please make it available without restrictions. Please ensure that the code is sufficiently well documented and reusable, and that your Data Statement in the Editorial Manager submission system accurately describes where your code can be found. 

(H) Please ensure that your Data Statement in the submission system accurately describes where your data can be found and is in final format, as it will be published as written there. 

We expect to receive your revised manuscript within three weeks. 

*Published Peer Review History*

*Press*

Best wishes,

Richard

Richard Hodge, PhD

rhodge@plos.org

Reviewer remarks:

Reviewer #1: In this manuscript, the authors explore phosphoproteomic changes in the fission yeast S. pombe during highly synchronous sexual differentiation, highlighting a significant role for TORC1. The authors have adequately addressed the previous review's comments, resulting in a much improved manuscript.

Reviewer #2: This manuscript has satisfied this reviewer's comments in their response. I think it is worthy of acceptance. That said, I believe that several minor changes to the text would be an improvement. These are not required for acceptance 

1.Reviewer comment on response to reviewer 1:

2 In Figure 5C, the phosphorylation level of Psk1 in the S1837E mutant is as low as in the h- DMSO 

treatment, but Rps6 is phosphorylated. Is this phosphorylation of Rps6 Psk1-independent, differing 

from the results in Figure 5D? 

This is a good question for which we can only speculate. The P-Rps6 signal is indeed increasing in 

the tor2-S1837E mutant whereas the P-Psk1 signal is not. P-Psk1 is a direct TORC1 target, 

whereas P-Rps6 is phosphorylated both downstream of Psk1 and downstream of TORC2-Gad8 

(and perhaps also other kinases). Furthermore, there are complex cross-talks between TORC1 and 

TORC2. Thus, interpretation of the P-Rps6 signal is more difficult than the P-Psk1 signal. One 

possibility is that constitutive low TORC1 activity in the tor2-S1837E mutant leads to cell adaptation 

through some compensatory effect in signaling - for instance increase in TORC2 activity - which 

leads to the observed P-Rps6 signal. Such adaptation would likely not happen in psk1∆ cells, 

because these cells have no known phenotype (and therefore little adaptation pressure). It would 

also not happen upon acute TORC1 inhibition with rapamycin, which leads to loss of P-Rps6 as 

shown in Figure 5B. 

This reviewer thinks the authors response here is on target and should be a part of the discussion. They only do one experiment to support the discussion and it might be something to raise as an outstanding question. 

2. This reviewer is commenting on the response below to reviewer 1:

3 The authors suggest that autophagy is not induced by pheromone signaling (page 10). However, 

in Figure 6B, the CFP signal, which is an autophagy marker, appears to be detected more strongly 

at 6 hours in the pheromone-treated samples compared to the control. 

We have repeated this experiment several times, and the possible higher amount of cleaved CFP 

was not reproducible. We have exchanged the blot shown in the first version of the manuscript for 

one more representative of our observations. We also show a second replicate of this result in 

Figure S7D. We note that previous literature also concluded that there is no induction of autophagy 

during mating (Nakashima et al, Curr Genet 2006; Kohda et al, Genes to Cell 2007). 

Author text carefully explains:

""We first used these conditions to probe whether starvation-dependent induction of autophagy is modulated by pheromone signaling. As a readout, we used CFP-Atg8 cleavage, as CFP-Atg8 is delivered to vacuoles by the autophagic process, where only the CFP moiety is resistant to vacuolar proteases. In WT cells, CFP-Atg8 cleavage occurred upon transfer to low-glutamate conditions and increased over time in a similar manner whether cells were treated with MetOH or P-factor (Fig 6B). This suggests that pheromone signaling does not induce autophagy. 

When we probed CFP-Atg8 cleavage in atg1∆ as control, we were surprised to observe significant residual levels of free CFP (Figure 6C, S7B), as deletion of this upstream regulatory kinase is reported to fully block autophagy (Mukaiyama et al., 2009; Pan et al., 2020; Sun et al., 2013). We reasoned that this difference may be due to differences in media or nitrogen source, as we used MSL + glutamate, while earlier studies used EMM + ammonium. Indeed, whereas exchanging glutamate for ammonium during growth before starvation had no effect on the observed cleavage in MSL medium (Figure S7B), using EMM medium abrogated cleavage (Figure 6C), as previously reported. Thus, for currently unknown reasons, deletion of Atg1 kinase only partially prevents autophagy in MSL medium. """

This reviewer thinks the discussion needs better summary relating to the differences with glutamate versus other nitrogen sources as well as listing of the use of glutamate instead of ammonium and the type of medium in their caveat paragraph. 

3. Request for changes to the caveat paragraph in Discussion: I paste the paragraph below and insert suggested changes as a guide to the authors in bold font with brackets so the authors can understand what this reviewer thinks needs to be included in the paragraph.

First, the cyc5∆ cells lacking all [FIVE] non-essential cyclins [i.e. cig1∆ cig2∆ crs1∆ puc1∆ rem1∆ ], which extends the differentiation-permissive G1 phase [and greatly increases cell length], may mask some of the

approach and suggest that identified phosphorylation events also occur in wildtype strains. Third, we used glutamate instead of ammonium to hasten differentiation which may affect the physiological response in unknown ways and discovered that the medium (i.e. MSL versus EMM) influences signaling outcome (see discussion of Figures 6 and S7B in results)."""

"""We have set up robust synchronization procedures to conduct population-based assays of mating and fusing cells, which will be of general use for other -omics analyses. It is however important to bear in mind the specific conditions allowing optimal synchrony. First, the cyc5∆ cells lacking all [FIVE] non-essential cyclins [i.e. cig1∆ cig2∆ crs1∆ puc1∆ rem1∆ ], which extends the differentiation-permissive G1 phase [and greatly increases cell length], may mask some of the (phospho-)proteomic changes governing the starvation- and/or pheromone-dependent cell cycle arrest of wildtype cells (Stern and Nurse, 1997; Stern and Nurse, 1998). Second, because of the use of the split fus1opto allele for synchronization during cell-cell fusion, we should be careful about interpretation of phosphorylation events on Fus1 and accessory proteins, as these may not represent native changes at the fusion site. Nevertheless, the few phosphorylation events that were expected (for instance that of the MAPK Spk1 active site; (Kelsall et al., 2019)) or independently confirmed here (Rps6) validate the approach and suggest that identified phosphorylation events also occur in wildtype strains. Third, we used glutamate instead of ammonium to hasten differentiation which may affect the physiological response in unknown ways and found evidence that the type of medium (i.e. MSL versus EMM) also may influence signaling outcome (see discussion of Figures 6 and S7B in results).""" 

3. With respect to discussion about phospho-proteomics presentation in Figure S5E. The author appropriately responded to reviewer comments. In response to author question about S5E cluster motifs, this reviewer thinks E could be enlarged to make it more readable rather than removed but this is not essential. 

4. With respect to the choice of subtracting starvation data to reveal mating specific data. As the authors correctly wrote in the first sentence of their abstract: "Starvation, which is associated with inactivation of the growth-promoting TOR complex 1 (TORC1), is a strong environmental signal for cell differentiation." 

The authors have the right to present the data as they wish and subtract out the starvation data. However, starvation induced events may be involved in some of the mating response which is induced under conditions of nitrogen starvation in the presence of cells of opposite mating type. In addition, in their phosphor-proteomic analysis they did not uncover all expected phosphorylation sites for mating which leads one to wonder if any were removed during their subtraction. If this was checked it should be stated in the results section or methods section. 

5. Figure S2B is a positive addition. A sentence that explains clearly that some but not all expected sites were found along with new ones should be included in its presentation. It is worth considering whether the missing ones found in the starvation phospho-proteomic data that was subtracted out and if not that might also be mentioned.

Reviewer #3: I have now gone through the manuscript and appreciate the efforts made to address my concerns. The authors provided additional data and clarifications that strengthen the manuscript and make it suitable for publication.

Of note, I understand why the authors feel uncomfortable about quantifying Western blots, but I think that one simple way to get an estimation is to load for example 3 lanes with decreasing quantities of one extract to give an idea of the range of signal variation that is detectable on a blot. Nevertheless, the key point of reproducibility was satisfactorily addressed.

---

## [Editor Report · Decision Letter 2]

2 Dec 2024

Dear Sophie,

On behalf of my colleagues and the Academic Editor, Sarah Zanders, I am pleased to say that we can in principle accept your manuscript for publication, provided you address any remaining formatting and reporting issues. These will be detailed in an email you should receive within 2-3 business days from our colleagues in the journal operations team. Please note that we will not be able to formally accept your manuscript and schedule it for publication until you have completed any requested changes.

*IMPORTANT*

In addition, there are a few additional points that we still need you to address but can be completed during the production process:

(A) I note that a table of gene targets may be missing in the 'Figure 4F-H' tab of the S1 Data File (similar to the table included in the 'Figure 3F-I' tab?).

(B) I completely forgot to mention in my previous decision letter that for papers from Editorial board members, it is now our policy to include a sentence in the conflict of interest statement saying something to the effect of ‘SGM is a member of the PLOS Biology Editorial Board’. I would be grateful if you could please include this sentence in the competing interests section in the online submission form during the production process. 

(C) Once the DOI is assigned for your publication, I would be grateful if you could please make the proteomic and phosphoproteomic datasets deposited in the PRIDE repository publicly available. 

PRESS

Best wishes,

Richard 

Richard Hodge, PhD

rhodge@plos.org

PLOS
